# RETHINKING THE UNIFORMITY METRIC IN SELF-SUPERVISED LEARNING

**Xianghong Fang**
The Chinese University of Hong Kong, Shenzhen
fangxianghong2@gmail.com

**Jian Li**
Tencent AI Lab
lijianjack@gmail.com

**Qiang Sun**[*]
University of Toronto & MBZUAI
qsunstats@gmail.com

**Benyou Wang**[*]
The Chinese University of Hong Kong, Shenzhen & SRIBD
wangbenyou@cuhk.edu.cn

## ABSTRACT

Uniformity plays an important role in evaluating learned representations, providing insights into self-supervised learning. In our quest for effective uniformity metrics, we pinpoint four principled properties that such metrics should possess. Namely, an effective uniformity metric should remain invariant to instance permutations and sample replications while accurately capturing feature redundancy and dimensional collapse. Surprisingly, we find that the uniformity metric proposed by Wang & Isola (2020) fails to satisfy the majority of these properties. Specifically, their metric is sensitive to sample replications, and can not account for feature redundancy and dimensional collapse correctly. To overcome these limitations, we introduce a new uniformity metric based on the Wasserstein distance, which satisfies all the aforementioned properties. Integrating this new metric in existing self-supervised learning methods effectively mitigates dimensional collapse and consistently improves their performance on downstream tasks involving CIFAR-10 and CIFAR-100 datasets. Code is available at https://github.com/statsle/WassersteinSSL.

## 1 INTRODUCTION

Self-supervised learning excels in acquiring invariant representations to various augmentations (Chen et al., 2020; He et al., 2020; Caron et al., 2020; Grill et al., 2020; Zbontar et al., 2021). It has been outstandingly successful across a wide range of domains, such as multimodality learning, object detection, and segmentation (Radford et al., 2021; Li et al., 2022; Xie et al., 2021; Wang et al., 2021; Yang et al., 2021; Zhao et al., 2021). To gain a deeper understanding of self-supervised learning, thoroughly evaluating the learned representations is necessary (Wang & Isola, 2020; Gao et al., 2021; Tian et al., 2021; Jing et al., 2022).

Alignment, a metric quantifying the similarities between positive pairs, holds significant importance in the evaluation of learned representations (Wang & Isola, 2020). It ensures that positive pairs are mapped to similar features, making them invariant to unnecessary details (Hadsell et al., 2006; Chen et al., 2020). However, relying solely on alignment proves inadequate for effectively assessing the representations. This limitation becomes evident in the presence of extremely small alignment values in collapsing solutions, as observed in Siamese networks (Hadsell et al., 2006), where all outputs collapse to a single point (Chen & He, 2021), as illustrated in Figure 1. In such cases, the learned representations exhibit optimal alignment but fail to provide meaningful information for any downstream tasks. This underscores the necessity of incorporating additional metrics when evaluating learned representations.

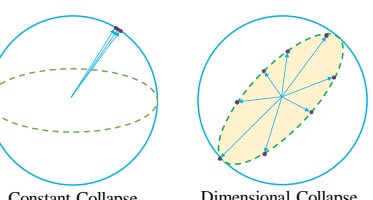

Constant Collapse          Dimensional Collapse

Figure 1: The left figure presents constant collapse, and the right figure visualizes dimensional collapse.

---

[*]Qiang Sun and Benyou Wang are joint corresponding authors.

To further evaluate the learned representations, Wang & Isola (2020) formally introduced a *uniformity* metric based on the logarithm of the average pairwise Gaussian potential (Cohn & Kumar, 2007). Uniformity assesses how feature embeddings are distributed uniformly across the unit hypersphere, and higher uniformity indicates more information from the data is preserved. Since its introduction, uniformity has played a pivotal role in understanding self-supervised learning and mitigating constant collapse (Arora et al., 2019; Wang & Isola, 2020; Gao et al., 2021). Nevertheless, the effectiveness of this particular uniformity metric warrants further examination.

To delve deeper into the existing uniformity metric proposed by Wang & Isola (2020), we introduce four principled properties that an effective uniformity metric should possess. Guided by these properties, we conduct a theoretical analysis, unveiling key limitations of this metric, particularly its inability to capture feature redundancy and dimensional collapse (Hua et al., 2021). Dimensional collapse refers to the scenario where representations occupy a lower-dimensional subspace rather than the entire embedding space (Jing et al., 2022); see Figure 1. We reinforce our theoretical findings with empirical evidence, demonstrating, for instance, the existing metric's inability to differentiate between different degrees of dimensional collapse. Subsequently, we propose a novel uniformity metric based on the quadratic Wasserstein distance that satisfies all four properties, thereby surpassing the existing one. Finally, integrating the proposed uniformity metric as an auxiliary loss within existing self-supervised learning methods consistently enhances their performance in downstream tasks.

Our main contributions are summarized as follows. (i) We identify four principled properties that an effective uniformity metric should possess, providing new guidelines on designing such metrics. (ii) Surprisingly, we find that the existing uniformity metric (Wang & Isola, 2020) fails to meet the majority of these properties. For example, it can not correctly capture dimensional collapse. (iii) We propose a new uniformity metric based on the Wasserstein distance that satisfies all four properties, addressing key limitations of the existing metric. (iv) Our proposed uniformity metric can seamlessly integrate as an auxiliary loss in various self-supervised learning methods, resulting in improved performance in downstream tasks.

## 2 BACKGROUND

### 2.1 SELF-SUPERVISED REPRESENTATION LEARNING

Self-supervised learning leverages the idea that similar samples should have similar representations that are invariant to unnecessary details (Wang & Isola, 2020). For instance, the Siamese network (Hadsell et al., 2006) takes as input positive pairs $(\mathbf{x}^a, \mathbf{x}^b)$, often obtained by taking two augmented views of the same sample $\mathbf{x}$. These positive pairs are then processed by an encoder network $f$ consisting of a backbone (e.g., ResNet (He et al., 2016)) and a projection MLP head (Chen et al., 2020), yielding representations $(\mathbf{z}^a = f(\mathbf{x}^a), \mathbf{z}^b = f(\mathbf{x}^b))$[1]. To enforce invariance, a natural approach is to minimize the following alignment loss, defined as the expected distance between positive pairs:

$$\mathcal{L}_{\mathcal{A}} := \mathbb{E}_{(\mathbf{z}^a, \mathbf{z}^b) \sim p_{\text{pos}}} \left\| \mathbf{z}_i^a - \mathbf{z}_i^b \right\|_2^2, \tag{1}$$

where $p_{\text{pos}}(\cdot, \cdot)$ is the distribution of positive pairs.

However, optimizing the above alignment loss alone may lead to an undesired collapsing solution, where all representations collapse into a single point, as shown in Figure 1.

### 2.2 EXISTING SOLUTIONS TO CONSTANT COLLAPSE

To prevent constant collapse, existing solutions include contrastive learning, asymmetric model architecture, and redundancy reduction.

**Contrastive Learning**   Contrastive learning offers a potent solution to mitigate constant collapse. The key idea is to leverage negative pairs. For example, SimCLR (Chen et al., 2020) introduced an in-batch negative sampling strategy that utilizes samples within a batch as negative samples. However, its effectiveness is contingent on the use of a large batch size. To address this limitation,

---

[1]For simplicity, we also refer to $(\mathbf{z}^a, \mathbf{z}^b)$ as positive pairs.

MoCo (He et al., 2020) used a memory bank, which stores additional representations as negative samples. Recent research endeavors have also explored clustering-based contrastive learning, which combines a clustering objective with contrastive learning techniques (Li et al., 2021; Caron et al., 2020).

**Asymmetric Model Architecture**   The use of asymmetric model architecture represents another strategy to combat constant collapse. One plausible explanation for its effectiveness is that such an asymmetric design encourages encoding more information (Grill et al., 2020). To maintain this asymmetry, BYOL (Grill et al., 2020) introduces the concept of using an additional predictor in one branch of the Siamese network while employing momentum updates and stop-gradient operators in the other branch. DINO (Caron et al., 2021), takes this asymmetry a step further by applying it to two encoders, distilling knowledge from the momentum encoder into the other one (Hinton et al., 2015). SimSiam (Chen & He, 2021) removes the momentum update from BYOL, and shows that the momentum update may not be essential in preventing constant collapse. However, Mirror-SimSiam (Zhang et al., 2022a) swaps the stop-gradient operator to the other branch. Its failure challenges the assertion made in SimSiam (Chen & He, 2021) that the stop-gradient operator is the key component for preventing constant collapse. Tian et al. (2021) provides a theoretical examination to elucidate why an asymmetric model architecture can effectively avoid constant collapse.

**Redundancy Reduction**   The fundamental principle behind redundancy reduction to mitigate constant collapse is to maximize the information preserved by the representations. The key idea is to decorrelate the learned representations. Barlow Twins (Zbontar et al., 2021) aims to achieve decorrelation by focusing on the cross-correlation matrix, while VICReg (Bardes et al., 2022) focuses on the covariance matrix. Zero-CL (Zhang et al., 2022b) takes a hybrid approach, combining instance-wise and feature-wise whitening techniques.

## 2.3   THE EXISTING UNIFORMITY METRIC

While the aforementioned solutions effectively prevent constant collapse, they are not as effective in preventing dimensional collapse, wherein representations occupy a lower-dimensional subspace instead of the entire space. This phenomenon has been observed in contrastive learning by visualizing the singular value spectra of representations (Jing et al., 2022; Tian et al., 2021).

To quantitatively measure the degree of collapse, Wang & Isola (2020) introduced a uniformity loss based on the logarithm of the average pairwise Gaussian potential. Given (normalized) feature representations $\{\mathbf{z}_1, \mathbf{z}_2, ..., \mathbf{z}_n\}$, their proposed empirical uniformity loss is:

$$\mathcal{L}_{\mathcal{U}} := \log \frac{1}{n(n-1)/2} \sum_{i=2}^{n} \sum_{j=1}^{i-1} e^{-t\|\mathbf{z}_i - \mathbf{z}_j\|_2^2}, \tag{2}$$

where $t > 0$ is a fixed parameter, often set to 2. Then $-\mathcal{L}_{\mathcal{U}}$ serves as the corresponding uniformity metric, with a higher value indicating greater uniformity.

We demonstrate in this work that this metric is insensitive to dimensional collapse, both theoretically in Section 3.2 and empirically in Section 5.2.

## 3   WHAT MAKES AN EFFECTIVE UNIFORMITY METRIC?

In this section, we begin by presenting four fundamental properties that an effective uniformity metric should possess. Leveraging these properties as a lens, we then scrutinize the existing uniformity metric $-\mathcal{L}_{\mathcal{U}}$, shedding light on its limitations.

### 3.1   FOUR PROPERTIES FOR UNIFORMITY

A uniformity metric $\mathcal{U} : \mathbb{R}^{mn} \to \mathbb{R}$ is a function that maps a set of learned representations to a scalar indicator of uniformity. In the following section, we introduce four principled properties that an effective uniformity metric should possess. Let $\mathcal{D} = \mathbf{z}_1, \ldots, \mathbf{z}_n \in \mathbb{R}^{mn}$ represent the learned representations. To avoid the trivial case, we assume that $\mathbf{z}_1, \ldots, \mathbf{z}_n$ are not all equal, meaning that not all points collapse to a single constant point.

First, an effective uniformity metric should be invariant to the permutation of instances, as the distribution of representations should not be affected by permutations.

**Property 1** (Instance Permutation Constraint (IPC)). *An effective uniformity metric $\mathcal{U}$ should satisfy*

$$\mathcal{U}(\pi(\mathcal{D})) = \mathcal{U}(\mathcal{D}), \tag{3}$$

*where $\pi$ is a permutation over the instances.*

Second, an effective uniformity metric should be invariant to instance clones, as instance cloning does not vary the distribution of representations.

**Property 2** (Instance Cloning Constraint (ICC)). *An effective uniformity metric $\mathcal{U}$ should satisfy*

$$\mathcal{U}(\mathcal{D} \uplus \mathcal{D}) = \mathcal{U}(\mathcal{D}), \tag{4}$$

*where $\mathcal{D} \uplus \mathcal{D} := \{\mathbf{z}_1, \mathbf{z}_2, ..., \mathbf{z}_n, \mathbf{z}_1, \mathbf{z}_2, ..., \mathbf{z}_n\}$.*

Third, an effective uniformity metric should strictly decrease as feature-level cloning for each instance occurs, as this duplication introduces redundancy, which corresponds to dimensional collapse (Zbontar et al., 2021; Bardes et al., 2022).

**Property 3** (Feature Cloning Constraint (FCC)). *An effective uniformity metric $\mathcal{U}$ should satisfy*

$$\mathcal{U}(\mathcal{D} \oplus \mathcal{D}) < \mathcal{U}(\mathcal{D}), \tag{5}$$

*where $\mathcal{D} \oplus \mathcal{D} := \{\mathbf{z}_1 \oplus \mathbf{z}_1, \mathbf{z}_2 \oplus \mathbf{z}_2, ..., \mathbf{z}_n \oplus \mathbf{z}_n\}$ and $\mathbf{z}_i \oplus \mathbf{z}_i := (z_{i1}, \cdots, z_{im}, z_{i1}, \cdots, z_{im})^{\mathrm{T}} \in \mathbb{R}^{2m}$.*

Fourth, an effective uniformity metric should strictly decrease with the addition of constant features for each instance, as this introduces uninformative and thus redundant features, which again corresponds to dimensional collapse.

**Property 4** (Feature Baby Constraint (FBC)). *An effective uniformity metric $\mathcal{U}$ should satisfy*

$$\mathcal{U}(\mathcal{D} \oplus \mathbf{0}^k) < \mathcal{U}(\mathcal{D}), \quad k \in \mathbb{N}^+, \tag{6}$$

*where $\oplus$ is defined in Property 3, that is, $\mathcal{D} \oplus \mathbf{0}^k = \{\mathbf{z}_1 \oplus \mathbf{0}^k, \mathbf{z}_2 \oplus \mathbf{0}^k, ..., \mathbf{z}_n \oplus \mathbf{0}^k\}$ and $\mathbf{z}_i \oplus \mathbf{0}^k = (z_{i1}, z_{i2}, ..., z_{im}, 0, 0, ..., 0)^{\mathrm{T}} \in \mathbb{R}^{m+k}$.*

Intuitively, Properties 1 and 2 ensure that the uniformity metric should remain insensitive to instance permutations and sample replications, respectively. Meanwhile, Properties 3 and 4 ensure that feature redundancy and dimensional collapse reduce the uniformity metric, as they make the distribution of the representations less uniform. These four properties constitute intuitive yet principled characteristics of an effective uniformity metric.

## 3.2 Examining the uniformity metric $-\mathcal{L}_\mathcal{U}$

We employ the four properties introduced earlier to analyze the uniformity metric $-\mathcal{L}_\mathcal{U}$ defined in Eqn. (2). The following theorem summarizes our findings.

**Theorem 1.** *The uniformity metric $-\mathcal{L}_\mathcal{U}$ satisfies Property 1, but violates Properties 2, 3, and 4.*

The proof of the above theorem is provided in Appendix C. The violation of Property 2 indicates that the uniformity metric $-\mathcal{L}_\mathcal{U}$ is sensitive to sample replications, while the violations of Properties 3 and 4 suggest that feature redundancy and dimensional collapse do not reduce the uniformity metric $-\mathcal{L}_\mathcal{U}$, making this uniformity metric unable to correctly reflect feature redundancy and dimensional collapse. Therefore, there is a pressing need to develop a new uniformity metric.

## 4 A New Uniformity Metric

In this section, we introduce a new uniformity metric to address the limitations of $-\mathcal{L}_\mathcal{U}$.

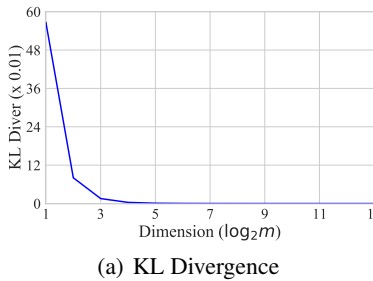
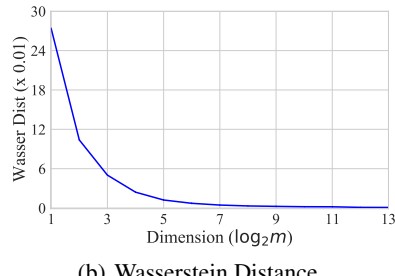

(a) KL Divergence

(b) Wasserstein Distance

Figure 2: The KL divergence and Wasserstein distance between $Y_i$ and $\widehat{Y}_i$ w.r.t. various dimensions.

### 4.1 THE UNIFORM SPHERICAL DISTRIBUTION AND AN APPROXIMATION

As pointed out by (Wang & Isola, 2020), feature vectors should be roughly uniformly distributed on the unit hypersphere $\mathcal{S}^{m-1}$, preserving as much information of the data as possible. Therefore, we adopt the uniform spherical distribution as our target distribution.

Our approach utilizes the quadratic Wasserstein distance, a form of statistical distance, between the feature distribution and the target distribution as the new uniformity loss. However, computing any statistical distances involving the uniform spherical distribution can be challenging. To address this, we first establish an asymptotic equivalence between the uniform spherical distribution and the isotropic Gaussian distribution. By adopting a Gaussian distribution for the representations, we then exploit the fact that the quadratic Wasserstein distance between two Gaussian distributions has a closed form involving only the means and covariance matrices, leading to a new and simple uniformity loss. We need the following fact.

**Fact 1.** *If* $\mathbf{Z} \sim \mathcal{N}(\mathbf{0}, \sigma^2 \mathbf{I}_m)$*, then* $\mathbf{Y} := \mathbf{Z}/\|\mathbf{Z}\|_2$ *is uniformly distributed on the unit hypersphere* $\mathcal{S}^{m-1}$.

Because the average length of $\|\mathbf{Z}\|_2$ is roughly $\sigma\sqrt{m}$ (Chandrasekaran et al., 2012), that is,

$$\frac{m}{\sqrt{m+1}} \leq \|\mathbf{Z}\|_2/\sigma \leq \sqrt{m},$$

we expect that $\mathbf{Z}/(\sigma\sqrt{m}) \sim \mathcal{N}(\mathbf{0}, \mathbf{I}_m/m)$ provides a reasonable approximation to $\mathbf{Z}/\|\mathbf{Z}\|_2$, and thus to the uniform spherical distribution. This is partially justified by the following theorem.

**Theorem 2.** *Let* $Y_i$ *be the* $i$-th *coordinate of* $\mathbf{Y} = \mathbf{Z}/\|\mathbf{Z}\|_2 \in \mathbb{R}^m$*, where* $\mathbf{Z} \sim \mathcal{N}(\mathbf{0}, \sigma^2 \mathbf{I}_m)$*. Then the quadratic Wasserstein distance between* $Y_i$ *and* $\widehat{Y}_i \sim \mathcal{N}(0, 1/m)$ *converges to zero as* $m \to \infty$*, that is,*

$$\lim_{m \to \infty} \mathcal{W}_2(Y_i, \widehat{Y}_i) = 0.$$

Theorem 2 suggests that $\mathcal{N}(\mathbf{0}, \mathbf{I}_m/m)$ approximates the distribution of each coordinate of the uniform spherical distribution as $m \to \infty$. It can be proven by first employing the Talagrand $T_2$ inequality (Van Handel, 2016) to upper bound the quadratic Wasserstein distance using the Kullback-Leibler (KL) divergence, and then establishing that the Kullback-Leibler (KL) divergence converges to 0. The proof is provided in Appendix B.

We empirically compare the distributions of $Y_i$ and $\widehat{Y}_i$ across various dimensions $m \in 2, 4, 8, 16, 32, 64, 128, 256$. For each $m$, we sample 200,000 data points from both $Y_i$ and $\widehat{Y}_i$, bin them into 51 groups, and calculate the empirical KL divergence and Wasserstein distance. Figure 2 plots both distances versus increasing dimensions. We observe that both distances converge to 0 as $m$ increases. Specifically, these results indicate that the distribution of $\widehat{Y}_i$ provides a reasonable approximation to that of $Y_i$ when $m \geq 2^4 = 16$. Further comparisons between $\mathbf{Y}$ and $\widehat{\mathbf{Y}}$ can be found in Appendix D.

### 4.2 A NEW METRIC FOR UNIFORMITY

In this section, we discuss how to use the quadratic Wasserstein distance between the distribution of learned representations and $\mathcal{N}(\mathbf{0}, \mathbf{I}_m/m)$, in place of the uniform spherical distribution $\text{Unif}(\mathcal{S}^{m-1})$, as our new uniformity loss.

To facilitate computation, we adopt a Gaussian hypothesis for the learned representations and assume they follow $\mathcal{N}(\boldsymbol{\mu}, \boldsymbol{\Sigma})$. With this assumption, we employ the quadratic Wasserstein distance[2] to measure the distance between two distributions. We need the following well-known lemma (Olkin & Pukelsheim, 1982).

**Lemma 1.** *Then the quadratic Wasserstein distance between $\mathcal{N}(\boldsymbol{\mu}, \boldsymbol{\Sigma})$ and $\mathcal{N}(\mathbf{0}, \mathbf{I}/m)$ is*

$$\sqrt{\|\boldsymbol{\mu}\|_2^2 + 1 + \operatorname{tr}(\boldsymbol{\Sigma}) - \frac{2}{\sqrt{m}} \operatorname{tr}(\boldsymbol{\Sigma}^{\frac{1}{2}})}. \tag{7}$$

The lemma above indicates that the quadratic Wasserstein distance can be easily computed using the population mean and covariance of the representations. In practice, we estimate the population mean and covariance by using the sample mean $\widehat{\boldsymbol{\mu}}$ and covariance matrix $\widehat{\boldsymbol{\Sigma}}$, respectively. Specifically, the empirical quadratic Wasserstein distance serves as the new empirical uniformity loss:

$$\mathcal{W}_2 := \sqrt{\|\widehat{\boldsymbol{\mu}}\|_2^2 + 1 + \operatorname{tr}(\widehat{\boldsymbol{\Sigma}}) - \frac{2}{\sqrt{m}} \operatorname{tr}(\widehat{\boldsymbol{\Sigma}}^{\frac{1}{2}})}. \tag{8}$$

Thus, $-\mathcal{W}_2$ can be utilized as the new uniformity metric, with larger values indicating greater uniformity. Moreover, our new uniformity loss can be seamlessly integrated into various existing self-supervised learning methods to enhance their performance.

## 5 COMPARING TWO METRICS

### 5.1 THEORETICAL COMPARISON

We examine the proposed metric $-\mathcal{W}_2$ in terms of the four properties introduced earlier. The following theorem summarizes our findings.

**Theorem 3.** *The uniformity metric $-\mathcal{W}_2$ satisfies all four properties, that is, Properties 1–4.*

The proof of the above theorem is similar to that of Theorem 1, and is provided in Appendix C.2. Table 1 compares $-\mathcal{L}_{\mathcal{U}}$ and $-\mathcal{W}_2$. It is important to highlight that our new uniformity metric is invariant to instance permutations and sample replications, while effectively capturing feature redundancy and dimensional collapse.

Taking dimensional collapse as an example, we consider $\mathcal{D} \oplus \mathbf{0}^k$ versus $\mathcal{D}$. Here, a larger $k$ indicates a more severe dimensional collapse. However, $-\mathcal{L}_{\mathcal{U}}$ fails to identify this issue, as $-\mathcal{L}_{\mathcal{U}}(\mathcal{D} \oplus \mathbf{0}^k) = -\mathcal{L}_{\mathcal{U}}(\mathcal{D})$. In stark contrast, our proposed metric can accurately detect this dimensional collapse, as $-\mathcal{W}_2(\mathcal{D} \oplus \mathbf{0}^k) < -\mathcal{W}_2(\mathcal{D})$.

Table 1: Comparing the two uniformity metrics.

| Properties | IPC | ICC | FCC | FBC |
|---|---|---|---|---|
| $-\mathcal{L}_{\mathcal{U}}$ | ✔ | ✗ | ✗ | ✗ |
| $-\mathcal{W}_2$ | ✔ | ✔ | ✔ | ✔ |

### 5.2 EMPIRICAL COMPARISONS VIA SYNTHETIC STUDIES

We perform synthetic experiments to investigate the two uniformity metrics. An empirical examination of the correlation between these metrics shows that data points following an isotropic Gaussian distribution exhibit better uniformity compared to those from other distributions; see Appendix E for detailed results. Additionally, we generate data vectors from this distribution to enable a thorough comparison between the two metrics.

**On Dimensional Collapse Degrees** To generate data reflecting varying degrees of dimensional collapse, we sample data vectors from an isotropic Gaussian distribution, normalize them to have $\ell_2$ norms[3], and then zero out a proportion of the coordinates. As the proportion of zero-value coordinates, denoted by $\eta$, increases, dimensional collapse becomes more pronounced, while the proportion of

---

[2]We discuss using other statistical distances as uniformity losses, such as the Kullback-Leibler divergence and Bhattacharyya distance, in Appendix A.

[3]In this paper, we always first normalize the representations to have unit $\ell_2$ norms.

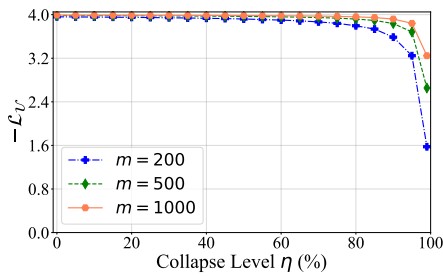 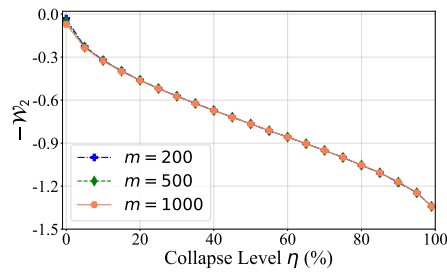

(a) Sensitivity to collapse degrees for $-\mathcal{L}_\mathcal{U}$  (b) Sensitivity to collapse degrees for $-\mathcal{W}_2$

Figure 3: Sensitivity to dimensional collapse degrees: $-\mathcal{W}_2$ is more sensitive than $-\mathcal{L}_\mathcal{U}$.

non-zero coordinates is $1 - \eta$. In Figure 3(a) and Figure 3(b), we observe that $-\mathcal{W}_2$ effectively captures different collapse degrees, whereas $-\mathcal{L}_\mathcal{U}$ remains almost unchanged even with $80\%$ collapse ($\eta = 80\%$), indicating that $-\mathcal{L}_\mathcal{U}$ is insensitive to the degrees of dimensional collapse.

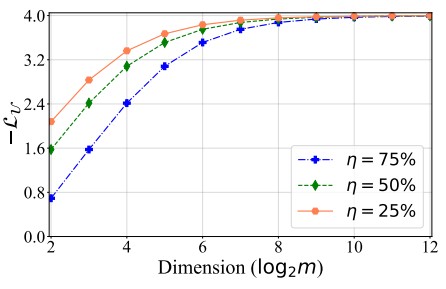 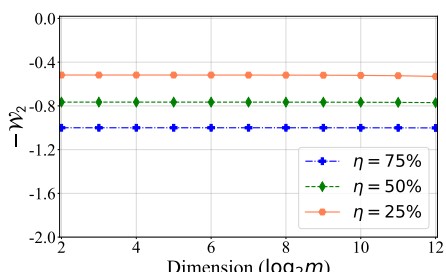

(a) Effectiveness of $-\mathcal{L}_\mathcal{U}$ when increasing $m$  (b) Effectiveness of $-\mathcal{W}_2$ of when increasing $m$

Figure 4: Effectiveness of the metrics when increasing dimension $m$: $-\mathcal{L}_\mathcal{U}$ fails to distinguish different dimensional collapse degrees for large $m$, while $-\mathcal{W}_2$ is always able to.

**On Sensitiveness of Dimensions** Figure 4 demonstrates that $-\mathcal{L}_\mathcal{U}$ can not distinguish between different degrees of dimensional collapse ($\eta = 25\%, 50\%$, and $75\%$) as the dimension $m$ increases (e.g., $m \geq 2^8 = 256$). In contrast, $-\mathcal{W}_2$ only depends on the degree of dimensional collapse and is independent of the dimensions $m$.

To complement the theoretical comparisons between the two metrics discussed in Section 5.1, we also conduct empirical comparisons in terms of FCC and FBC. ICC comparisons are collected in Appendix E.

**On Feature Cloning Constraint** We investigate the impact of feature cloning by creating multiple feature clones of the dataset, such as $\mathcal{D} \oplus \mathcal{D}$ and $\mathcal{D} \oplus \mathcal{D} \oplus \mathcal{D}$, corresponding to one and two times cloning, respectively. Figure 5(a) demonstrates that the value of $-\mathcal{L}_\mathcal{U}$ increases as the number of clones increases, which violates the strict decline in Eqn. (5). In contrast, in Figure 5(b), our proposed metric $-\mathcal{W}_2$ decreases, satisfying the property.

**On Feature Baby Constraint** We proceed to analyze the effect of feature baby, where we insert $k$ dimensional zero vectors into each instance of $\mathcal{D}$. This modified dataset is denoted as $\mathcal{D} \oplus \mathbf{0}^k$, and we examine the impact of $k$ on both metrics. Figure 6(a) shows that the value of $-\mathcal{L}_\mathcal{U}$ remains constant as $k$ increases, violating the strict inequality constraint in Eqn. (6). In contrast, Figure 6(b) shows that our proposed metric $-\mathcal{W}_2$ decreases, satisfying the constraint.

**Summary of Synthetic Studies** In summary, our empirical results corroborate our theoretical analysis, confirming that our proposed metric $-\mathcal{W}_2$ outperforms the existing metric $-\mathcal{L}_\mathcal{U}$ in capturing feature redundancy and dimensional collapse.

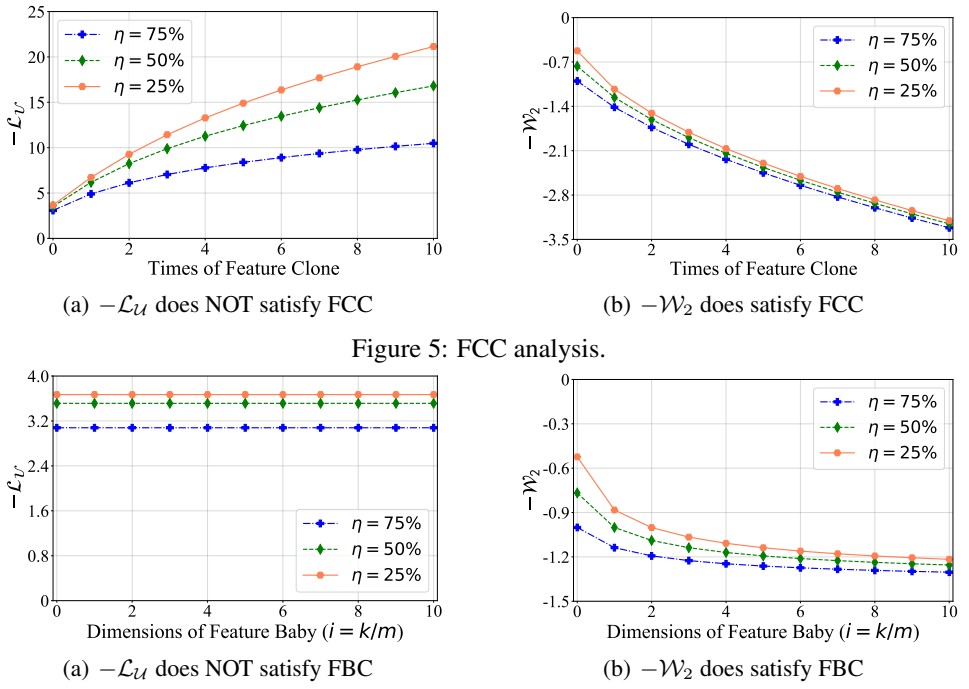

Figure 5: FCC analysis.

(a) $-\mathcal{L}_{\mathcal{U}}$ does NOT satisfy FBC

(b) $-\mathcal{W}_2$ does satisfy FBC

Figure 6: FBC analysis.

## 6 EXPERIMENTS

In this section, we integrate the proposed uniformity loss as an auxiliary term into various existing self-supervised methods. We then conduct experiments on CIFAR-10 and CIFAR-100 datasets to demonstrate its effectiveness.

**Models** We conduct experiments on a series of self-supervised representation learning models: (i) AlignUniform (Wang & Isola, 2020), which incorporates both alignment and uniformity losses in its objective function; (ii) three contrastive learning methods, namely SimCLR (Chen et al., 2020), MoCo (He et al., 2020), and NNCLR (Dwibedi et al., 2021); (iii) two asymmetric models, BYOL (Grill et al., 2020) and SimSiam (Chen & He, 2021); (iv) two methods based on redundancy reduction, BarlowTwins (Zbontar et al., 2021) and Zero-CL (Zhang et al., 2022b). To investigate the behavior of the proposed Wasserstein uniformity loss in self-supervised learning, we integrate it as an auxiliary loss into the following models: MoCo v2, BYOL, BarlowTwins, and Zero-CL. Additionally, we propose using linear decay to weight the Wasserstein uniformity loss during training. This is achieved by setting $\alpha_t = \alpha_{\max} - t, (\alpha_{\max} - \alpha_{\min})/T$, where $t$, $T$, $\alpha_{\max}$, $\alpha_{\min}$, and $\alpha_t$ represent the current epoch, maximum epochs, maximum weight, minimum weight, and current weight, respectively. Further details on the experimental settings can be found in Appendix F.1.

**Accuracy and representation capacity** We assess the aforementioned methods using two distinct criteria: accuracy and representation quality/capacity. Accuracy is gauged through linear evaluation accuracy, quantified by Top-1 accuracy (Acc@1) and Top-5 accuracy (Acc@5). On the other hand, representation quality/capacity is evaluated using the uniformity losses $\mathcal{L}_{\mathcal{U}}$ and $\mathcal{W}_2$, along with the alignment loss $\mathcal{L}_{\mathcal{A}}$. .

**Main Results** As depicted in Table 2, incorporating $\mathcal{W}_2$ as an additional loss consistently yields superior performance compared to models without this loss or those with $\mathcal{L}_{\mathcal{U}}$ as the additional term. Intriguingly, although it marginally compromises alignment, it enhances uniformity and accuracy in downstream tasks. This underscores the effectiveness of $\mathcal{W}_2$ as a uniformity loss. Notably, integrating the Wasserstein uniformity loss does not impede training or inference efficiency.

**Convergence Analysis** We evaluate the Top-1 accuracy of these models on CIFAR-10 and CIFAR-100 using the linear evaluation protocol, as described in Appendix F.2, across different training

Table 2: Main results on CIFAR-10 and CIFAR-100. Proj. and Pred. are the hidden dimensions in projector and predictor. ↑ and ↓ indicates gains and losses, respectively.

| Methods | Proj. | Pred. | CIFAR-10 | | | | | CIFAR-100 | | | | |
|---|---|---|---|---|---|---|---|---|---|---|---|---|
| | | | Acc@1↑ | Acc@5↑ | $\mathcal{W}_2$↓ | $\mathcal{L}_\mathcal{U}$↓ | $\mathcal{L}_\mathcal{A}$↓ | Acc@1↑ | Acc@5↑ | $\mathcal{W}_2$↓ | $\mathcal{L}_\mathcal{U}$↓ | $\mathcal{L}_\mathcal{A}$↓ |
| SimCLR | 256 | ✗ | 89.85 | 99.78 | 1.04 | -3.75 | 0.47 | 63.43 | 88.97 | 1.05 | -3.75 | 0.50 |
| NNCLR | 256 | 256 | 87.46 | 99.63 | 1.23 | -3.12 | 0.38 | 54.90 | 83.81 | 1.23 | -3.18 | 0.43 |
| SimSiam | 256 | 256 | 86.71 | 99.67 | 1.19 | -3.33 | 0.39 | 56.10 | 84.34 | 1.21 | -3.29 | 0.42 |
| AlignUniform | 256 | ✗ | 90.37 | 99.76 | 0.94 | -3.82 | 0.51 | 65.08 | 90.15 | 0.95 | -3.82 | 0.53 |
| MoCo v2 | 256 | ✗ | 90.65 | 99.81 | 1.06 | -3.75 | 0.51 | 60.27 | 86.29 | 1.07 | -3.60 | 0.46 |
| MoCo v2 + $\mathcal{L}_\mathcal{U}$ | 256 | ✗ | 90.98 ↑$_{0.33}$ | 99.67 | 0.98 ↑$_{0.08}$ | -3.82 | 0.53 ↓$_{0.02}$ | 61.21 ↑$_{0.94}$ | 87.32 | 0.98 ↑$_{0.09}$ | -3.81 | 0.52 ↓$_{0.06}$ |
| MoCo v2 + $\mathcal{W}_2$ | 256 | ✗ | 91.41 ↑$_{0.76}$ | 99.68 | 0.33 ↑$_{0.73}$ | -3.84 | 0.63 ↓$_{0.12}$ | 63.68 ↑$_{3.41}$ | 88.48 | 0.28 ↑$_{0.79}$ | -3.86 | 0.66 ↓$_{0.20}$ |
| BYOL | 256 | 256 | 89.53 | 99.71 | 1.21 | -2.99 | **0.31** | 63.66 | 88.81 | 1.20 | -2.87 | **0.33** |
| BYOL + $\mathcal{L}_\mathcal{U}$ | 256 | ✗ | 90.09 ↑$_{0.56}$ | 99.75 | 1.09 ↑$_{0.12}$ | -3.66 | 0.40 ↓$_{0.09}$ | 62.68 ↓$_{0.98}$ | 88.44 | 1.08 ↑$_{0.12}$ | -3.70 | 0.51 ↓$_{0.18}$ |
| BYOL + $\mathcal{W}_2$ | 256 | 256 | 90.31 ↑$_{0.78}$ | 99.77 | 0.38 ↑$_{0.83}$ | -3.90 | 0.65 ↓$_{0.34}$ | 65.16 ↑$_{1.50}$ | 89.25 | 0.36 ↑$_{0.84}$ | -3.91 | 0.69 ↓$_{0.36}$ |
| BarlowTwins | 256 | ✗ | 91.16 | 99.80 | 0.22 | -3.91 | 0.75 | 68.19 | 90.64 | 0.23 | -3.91 | 0.75 |
| BarlowTwins + $\mathcal{L}_\mathcal{U}$ | 256 | ✗ | 91.38 ↑$_{0.22}$ | 99.77 | 0.21 ↑$_{0.01}$ | -3.92 | 0.76 ↓$_{0.01}$ | 68.41 ↑$_{0.22}$ | 90.99 | 0.22 ↑$_{0.01}$ | -3.91 | 0.76 ↓$_{0.01}$ |
| BarlowTwins + $\mathcal{W}_2$ | 256 | ✗ | **91.43** ↑$_{0.27}$ | 99.78 | 0.19 ↑$_{0.03}$ | -3.92 | 0.76 ↓$_{0.01}$ | 68.47 ↑$_{0.28}$ | 90.64 | 0.19 ↑$_{0.04}$ | -3.91 | 0.79 ↓$_{0.04}$ |
| Zero-CL | 256 | ✗ | 91.35 | 99.74 | 0.15 | **-3.94** | 0.70 | 68.50 | 90.97 | 0.15 | -3.93 | 0.75 |
| Zero-CL + $\mathcal{L}_\mathcal{U}$ | 256 | ✗ | 91.28 ↓$_{0.07}$ | 99.74 | 0.15 | **-3.94** | 0.72 ↓$_{0.02}$ | 68.44 ↓$_{0.06}$ | 90.91 | 0.15 | -3.93 | 0.74 ↑$_{0.01}$ |
| Zero-CL + $\mathcal{W}_2$ | 256 | ✗ | 91.42 ↑$_{0.07}$ | **99.82** | **0.14** ↑$_{0.01}$ | **-3.94** | 0.71 ↓$_{0.01}$ | **68.55** ↑$_{0.05}$ | **91.02** | **0.14** ↑$_{0.01}$ | **-3.94** | 0.76 ↓$_{0.01}$ |

epochs. Figure 15 illustrates the results. By incorporating $\mathcal{W}_2$ as an additional loss for these models, we observe faster convergence compared to the raw models, particularly for MoCo v2 and BYOL, which exhibit significant collapse issues. Our experiments demonstrate that imposing the proposed Wasserstein uniformity metric as an auxiliary penalty loss greatly enhances uniformity but may compromise alignment. We further analyze uniformity and alignment throughout all training epochs in Appendix F.3.

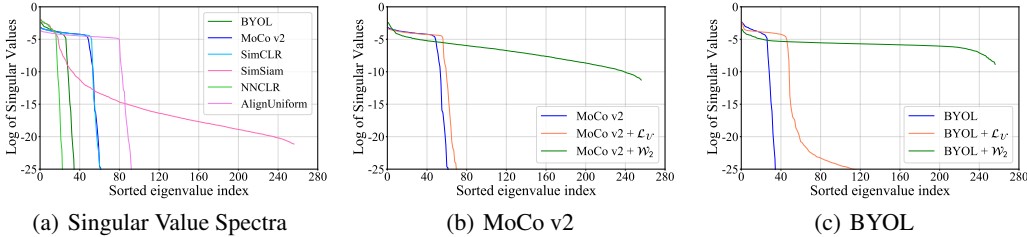

(a) Singular Value Spectra      (b) MoCo v2      (c) BYOL

Figure 7: Dimensional collapse analysis on CIFAR-100 dataset.

**Dimensional Collapse Analysis** We visualize the singular value spectra of the learned representations (Jing et al., 2022) for various models. These spectra contain the singular values of the covariance matrix of representations from CIFAR-100 dataset, sorted in logarithmic scale order. As shown in Figure 7(a), most singular values collapse to zeros in most models, indicating a large number of collapsed coordinates in most models. To further understand how the additional loss $\mathcal{W}_2$ helps prevent dimensional collapse, we add $\mathcal{W}_2$ as an additional loss for Moco v2 and BYOL, the numbers of collapsed coordinates decrease to zeros in both cases; see Figure 7(b) and Figure 7(c). This verifies that our proposed uniformity loss $\mathcal{W}_2$ can effectively address the dimensional collapse issue for Moco v2 and BYOL. In contrast, $\mathcal{L}_\mathcal{U}$ can not effectively prevent dimensional collapse.

## 7 CONCLUSION

In this paper, we have identified four principled properties that an effective uniformity metric should possess. Namely, an effective uniformity metric should remain invariant to instance permutations and sample replications while accurately capturing feature redundancy and dimensional collapse. Surprisingly, the popular uniformity metric proposed by Wang & Isola (2020) fails to meet the majority of these properties, unveiling its limitations. Empirical investigations corroborate our theoretical findings. To overcome these limitations, we introduce a new uniformity metric that satisfies all four properties. Particularly, this new metric demonstrates remarkable abilities to capture feature redundancy and dimensional collapse. Integrating it as an auxiliary loss in various self-supervised learning methods effectively mitigates dimensional collapse and consistently improves their performance on downstream tasks. Nonetheless, it is worth noting that the four identified properties may not encompass a comprehensive characterization of an ideal uniformity metric, warranting further exploration.

ACKNOWLEDGEMENT

Benyou Wang was partially supported by the Shenzhen Science and Technology Program (JCYJ20220818103001002), Shenzhen Doctoral Startup Funding (RCBS20221008093330065), and Tianyuan Fund for Mathematics of National Natural Science Foundation of China (NSFC) (12326608). Qiang Sun was partially supported in part by the Natural Sciences and Engineering Research Council of Canada under Grant RGPIN-2018-06484 and a Data Sciences Institute Catalyst Grant.

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

# Appendix

# Table of Contents

## A  STATISTICAL DISTANCES OVER GAUSSIAN DISTRIBUTIONS

We first introduce the Wasserstein distance or the earth mover distance.

**Definition 1.** The Wasserstein distance or earth-mover distance with $p$ norm is defined as below:

$$W_p(\mathbb{P}_r, \mathbb{P}_g) = (\inf_{\gamma \in \Pi(\mathbb{P}_r, \mathbb{P}_g)} \mathbb{E}_{(x,y)\sim\gamma}\big[\|x - y\|^p\big])^{1/p} . \tag{9}$$

where $\Pi(\mathbb{P}_r, \mathbb{P}_g)$ denotes the set of all joint distributions $\gamma(x, y)$ whose marginals are respectively $\mathbb{P}_r$ and $\mathbb{P}_g$. Intuitively, when viewing each distribution as a unit amount of earth/soil, the Wasserstein distance or earth-mover distance takes the minimum cost of transporting "mass" from $x$ to $y$ to transform the distribution $\mathbb{P}_r$ into the distribution $\mathbb{P}_g$. This distance is also called the quadratic Wasserstein distance when $p = 2$.

In this paper, we mainly exploit the quadratic Wasserstein distance over Gaussian distributions. Besides this distance, we also discuss other distribution distances as uniformity metrics and make comparisons with the Wasserstein distance. Specifically, the Kullback-Leibler divergence and the Bhattacharyya distance over Gaussian distributions are provided in Lemma 2 and Lemma 3 respectively. Both distances require full-rank covariance matrices, making them impropriate to conduct dimensional collapse analysis. In contrast, our quadratic Wasserstein distance-based uniformity metric is free of such a requirement.

**Lemma 2** (Kullback-Leibler divergence (Lindley & Kullback, 1959))**.** *Suppose two random variables* $\mathbf{Z}_1 \sim \mathcal{N}(\boldsymbol{\mu}_1, \boldsymbol{\Sigma}_1)$ *and* $\mathbf{Z}_2 \sim \mathcal{N}(\boldsymbol{\mu}_2, \boldsymbol{\Sigma}_2)$ *obey multivariate normal distributions, then Kullback-Leibler divergence between* $\mathbf{Z}1$ *and* $\mathbf{Z}_2$ *is:*

$$D_{\mathrm{KL}}(\mathbf{Z}_1, \mathbf{Z}_2) = \frac{1}{2}((\boldsymbol{\mu}_1 - \boldsymbol{\mu}_2)^T \boldsymbol{\Sigma}_2^{-1}(\boldsymbol{\mu}_1 - \boldsymbol{\mu}_2) + \mathrm{tr}(\boldsymbol{\Sigma}_2^{-1}\boldsymbol{\Sigma}_1 - \mathbf{I}) + \ln\frac{\det \boldsymbol{\Sigma}_2}{\det \boldsymbol{\Sigma}_1}).$$

**Lemma 3** (Bhattacharyya Distance (Bhattacharyya, 1943)). *Suppose two random variables $\mathbf{Z}_1 \sim \mathcal{N}(\boldsymbol{\mu}_1, \boldsymbol{\Sigma}_1)$ and $\mathbf{Z}_2 \sim \mathcal{N}(\boldsymbol{\mu}_2, \boldsymbol{\Sigma}_2)$ obey multivariate normal distributions, $\boldsymbol{\Sigma} = \frac{1}{2}(\boldsymbol{\Sigma}_1 + \boldsymbol{\Sigma}_2)$, then bhattacharyya distance between $\mathbf{Z}1$ and $\mathbf{Z}_2$ is:*

$$\mathcal{D}_B(\mathbf{Z}_1, \mathbf{Z}_2) = \frac{1}{8}(\boldsymbol{\mu}_1 - \boldsymbol{\mu}_2)^T \boldsymbol{\Sigma}^{-1}(\boldsymbol{\mu}_1 - \boldsymbol{\mu}_2) + \frac{1}{2}\ln\frac{\det\boldsymbol{\Sigma}}{\sqrt{\det\boldsymbol{\Sigma}_1\det\boldsymbol{\Sigma}_2}}.$$

## B  PROOF OF THEOREM 2

We first need the following lemma, whose proof is collected in the end of this section.

**Lemma 4.** *Let $\mathbf{Z} \sim \mathcal{N}(\mathbf{0}, \sigma^2\mathbf{I}_m)$ and $\mathbf{Y} = \mathbf{Z}/\|\mathbf{Z}\|_2$. Then the probability density function of $Y_i$, the $i$-th coordinate of $\mathbf{Y}$ is:*

$$f_{Y_i}(y_i) = \frac{\Gamma(m/2)}{\sqrt{\pi}\Gamma((m-1)/2)}(1 - y_i^2)^{(m-3)/2}, \ \ \forall\, y_i \in [-1, 1].$$

We are ready to prove Theorem 2.

*Proof of Theorem 2.* According to the Lemma 4, the pdf of $Y_i$ and $\widehat{Y}_i$ are:

$$f_{Y_i}(y) = \frac{\Gamma(m/2)}{\sqrt{\pi}\Gamma((m-1)/2)}(1 - y^2)^{(m-3)/2}, \quad f_{\widehat{Y}_i}(y) = \sqrt{\frac{m}{2\pi}}\exp\{-\frac{my^2}{2}\}.$$

Then the Kullback-Leibler divergence between $Y_i$ and $\widehat{Y}_i$ is

$$
\begin{aligned}
D_{\mathrm{KL}}(Y_i\|\widehat{Y}_i) &= \int_{-1}^{1} f_{Y_i}(y)[\log f_{Y_i}(y) - \log f_{\widehat{Y}_i}(y)]dy \\
&= \int_{-1}^{1} f_{Y_i}(y)[\log\frac{\Gamma(m/2)}{\sqrt{\pi}\Gamma((m-1)/2)} + \frac{m-3}{2}\log(1-y^2) - \log\sqrt{\frac{m}{2\pi}} + \frac{my^2}{2}]dy \\
&= \log\sqrt{\frac{2}{m}}\frac{\Gamma(m/2)}{\Gamma((m-1)/2)} + \int_{-1}^{1} f_{Y_i}(y)[\frac{m-3}{2}\log(1-y^2) + \frac{my^2}{2}]dy.
\end{aligned}
$$

Letting $\mu = y^2$, we have $y = \sqrt{\mu}$ and $dy = \frac{1}{2}\mu^{-\frac{1}{2}}du$. Thus,

$$
\begin{aligned}
\mathcal{A} &:= \int_{-1}^{1} f_{Y_i}(y)[\frac{m-3}{2}\log(1-y^2) + \frac{my^2}{2}]dy \\
&= 2\int_{0}^{1} \frac{\Gamma(m/2)}{\sqrt{\pi}\Gamma((m-1)/2)}(1-y^2)^{\frac{m-3}{2}}[\frac{m-3}{2}\log(1-y^2) + \frac{my^2}{2}]dy \\
&= \frac{\Gamma(m/2)}{\sqrt{\pi}\Gamma((m-1)/2)}\int_{0}^{1} (1-\mu)^{\frac{m-3}{2}}[\frac{m-3}{2}\log(1-\mu) + \frac{m}{2}\mu]\mu^{-\frac{1}{2}}d\mu \\
&= \frac{\Gamma(m/2)}{\sqrt{\pi}\Gamma((m-1)/2)}\frac{m-3}{2}\int_{0}^{1} (1-\mu)^{\frac{m-3}{2}}\mu^{-\frac{1}{2}}\log(1-\mu)d\mu \\
&\quad + \frac{\Gamma(m/2)}{\sqrt{\pi}\Gamma((m-1)/2)}\frac{m}{2}\int_{0}^{1} (1-\mu)^{\frac{m-3}{2}}\mu^{\frac{1}{2}}d\mu.
\end{aligned}
$$

By using the property of Beta distribution, and the inequality that $\frac{-\mu}{1-\mu} \leq \log(1-\mu) \leq -\mu$, we have

$$
\begin{aligned}
\mathcal{A}_1 &:= \frac{\Gamma(m/2)}{\sqrt{\pi}\Gamma((m-1)/2)} \frac{m-3}{2} \int_0^1 (1-\mu)^{\frac{m-3}{2}} \mu^{-\frac{1}{2}} \log(1-\mu) d\mu \\
&\leq -\frac{\Gamma(m/2)}{\sqrt{\pi}\Gamma((m-1)/2)} \frac{m-3}{2} \int_0^1 (1-\mu)^{\frac{m-3}{2}} \mu^{\frac{1}{2}} d\mu \\
&= -\frac{\Gamma(m/2)}{\sqrt{\pi}\Gamma((m-1)/2)} \frac{m-3}{2} B(\frac{3}{2}, \frac{m-1}{2}) \text{ and} \\
\mathcal{A}_2 &:= \frac{\Gamma(m/2)}{\sqrt{\pi}\Gamma((m-1)/2)} \frac{m}{2} \int_0^1 (1-\mu)^{\frac{m-3}{2}} \mu^{\frac{1}{2}} d\mu \\
&= \frac{\Gamma(m/2)}{\sqrt{\pi}\Gamma((m-1)/2)} \frac{m}{2} B(\frac{3}{2}, \frac{m-1}{2}).
\end{aligned}
$$

Then, for $\mathcal{A}$, we have

$$
\begin{aligned}
\mathcal{A} = \mathcal{A}_1 + \mathcal{A}_2 &\leq -\frac{\Gamma(m/2)}{\sqrt{\pi}\Gamma((m-1)/2)} \frac{m-3}{2} B(\frac{3}{2}, \frac{m-1}{2}) + \frac{\Gamma(m/2)}{\sqrt{\pi}\Gamma((m-1)/2)} \frac{m}{2} B(\frac{3}{2}, \frac{m-1}{2}) \\
&= \frac{3}{2} \frac{\Gamma(m/2)}{\sqrt{\pi}\Gamma((m-1)/2)} B(\frac{3}{2}, \frac{m-1}{2}) = \frac{3}{2} \frac{\Gamma(m/2)}{\sqrt{\pi}\Gamma((m-1)/2)} \frac{\Gamma(3/2)\Gamma((m-1)/2)}{\Gamma((m+2)/2)} \\
&= \frac{3}{2} \frac{\Gamma(3/2)\Gamma(m/2)}{\sqrt{\pi}\Gamma((m+2)/2)} = \frac{3}{2} \frac{(\sqrt{\pi}/2)\Gamma(m/2)}{\sqrt{\pi}\Gamma((m+2)/2)} = \frac{3}{4} \frac{\Gamma(m/2)}{\Gamma((m+2)/2)}.
\end{aligned}
$$

Using the Stirling formula, we have $\Gamma(x+\alpha) \to \Gamma(x)x^\alpha$ as $x \to \infty$ and thus

$$
\begin{aligned}
\lim_{m\to\infty} D_{\mathrm{KL}}(Y_i \| \widehat{Y}_i) &= \lim_{m\to\infty} \log \sqrt{\frac{2}{m}} \frac{\Gamma(m/2)}{\Gamma((m-1)/2)} + \lim_{m\to\infty} \mathcal{A} \\
&\leq \lim_{m\to\infty} \log \sqrt{\frac{2}{m}} \frac{\Gamma((m-1)/2)(\frac{m-1}{2})^{1/2}}{\Gamma((m-1)/2)} + \lim_{m\to\infty} \frac{3}{4} \frac{\Gamma(m/2)}{\Gamma((m+2)/2)} \\
&= \lim_{m\to\infty} \log \sqrt{\frac{2}{m}} \sqrt{\frac{m-1}{2}} + \frac{3}{4} \frac{\Gamma(m/2)}{\Gamma(m/2)m} = \lim_{m\to\infty} \log \sqrt{\frac{m-1}{m}} + \frac{3}{4m} = 0.
\end{aligned}
$$

We further use $T_2$ inequality (Van Handel, 2016, Theorem 4.31) to derive the quadratic Wasserstein metric (Van Handel, 2016, Definition 4.29) as:

$$
\lim_{m\to\infty} \mathcal{W}_2(Y_i, \widehat{Y}_i) \leq \lim_{m\to\infty} \sqrt{\frac{2}{m} D_{\mathrm{KL}}(Y_i \| \widehat{Y}_i)} = 0.
$$

$\square$

## B.1 PROOFS FOR SUPPORTING LEMMAS

*Proof of Lemma 4.* Let $\mathbf{Z} = [Z_1, Z_2, \cdots, Z_m] \sim \mathcal{N}(\mathbf{0}, \sigma^2 \mathbf{I}_m)$, then $Z_i \sim \mathcal{N}(0, \sigma^2), \forall i \in [1, m]$. Let $U = Z_i/\sigma \sim \mathcal{N}(0,1)$, $V = \sum_{j\neq i}^m (Z_j/\sigma)^2 \sim \mathcal{X}^2(m-1)$, then $U$ and $V$ are independent with each other. The random variable $T = \frac{U}{\sqrt{V/(m-1)}}$ follows the Student's t-distribution with $m-1$ degrees of freedom, and its probability density function (pdf) is:

$$
f_T(t) = \frac{\Gamma(m/2)}{\sqrt{(m-1)\pi}\Gamma((m-1)/2)} (1 + \frac{t^2}{m-1})^{-m/2}.
$$

For random variable $Y_i$, we have

$$
Y_i = \frac{Z_i}{\sqrt{\sum_{i=1}^m Z_i^2}} = \frac{Z_i}{\sqrt{Z_i^2 + \sum_{j\neq i}^m Z_j^2}} = \frac{Z_i/\sigma}{\sqrt{(Z_i/\sigma)^2 + \sum_{j\neq i}^m (Z_j/\sigma)^2}} = \frac{U}{\sqrt{U^2 + V}},
$$

and then $T = \frac{U}{\sqrt{V/(m-1)}} = \frac{\sqrt{m-1}Y_i}{\sqrt{1-Y_i^2}}$, $Y_i = \frac{T}{\sqrt{T^2+m-1}}$. Therefore, the cumulative distribution function (cdf) of $T$ is:

$$
\begin{aligned}
F_{Y_i}(y_i) = P(\{Y_i \leq y_i\}) &= \begin{cases} P(\{Y_i \leq y_i\}) & y_i \leq 0 \\ P(\{Y_i \leq 0\}) + P(\{0 < Y_i \leq y_i\}) & y_i > 0 \end{cases} \\
&= \begin{cases} P(\{\frac{T}{\sqrt{T^2+m-1}} \leq y_i\}) & y_i \leq 0 \\ P(\{\frac{T}{\sqrt{T^2+m-1}} \leq 0\}) + P(\{0 < \frac{T}{\sqrt{T^2+m-1}} \leq y_i\}) & y_i > 0 \end{cases} \\
&= \begin{cases} P(\{\frac{T^2}{T^2+m-1} \geq y_i^2, T \leq 0\}) & y_i \leq 0 \\ P(\{T \leq 0\} + P(\{\frac{T^2}{T^2+m-1} \leq y_i^2, T > 0\}) & y_i > 0 \end{cases} \\
&= \begin{cases} P(\{T \leq \frac{\sqrt{m-1}y_i}{\sqrt{1-y_i^2}}\}) & y_i \leq 0 \\ P(\{T \leq 0\} + P(\{0 < T \leq \frac{\sqrt{m-1}y_i}{\sqrt{1-y_i^2}}\}) & y_i > 0 \end{cases} \\
&= P(\{T \leq \frac{\sqrt{m-1}y_i}{\sqrt{1-y_i^2}}\}) = F_T(\frac{\sqrt{m-1}y_i}{\sqrt{1-y_i^2}}).
\end{aligned}
$$

The probability density function of $Y_i$ can then be derived as:

$$
\begin{aligned}
f_{Y_i}(y_i) = \frac{d}{dy_i} F_{Y_i}(y_i) &= \frac{d}{dy_i} F_T(\frac{\sqrt{m-1}y_i}{\sqrt{1-y_i^2}}) \\
&= f_T(\frac{\sqrt{m-1}y_i}{\sqrt{1-y_i^2}}) \frac{d}{dy_i}(\frac{\sqrt{m-1}y_i}{\sqrt{1-y_i^2}}) \\
&= [\frac{\Gamma(m/2)}{\sqrt{(m-1)\pi}\Gamma((m-1)/2)}(1-y_i^2)^{m/2}][\sqrt{m-1}(1-y_i^2)^{-3/2}] \\
&= \frac{\Gamma(m/2)}{\sqrt{\pi}\Gamma((m-1)/2)}(1-y_i^2)^{(m-3)/2}.
\end{aligned}
$$

$\square$

## C    EXAMINING THE FOUR PROPERTIES FOR TWO UNIFORMITY METRICS

### C.1    PROOF OF THEOREM 1: EXAMINING THE FOUR PROPERTIES FOR $-\mathcal{L}_{\mathcal{U}}$

Property 1 can be easily verified for $-\mathcal{L}_{\mathcal{U}}$ and thus we skip the verification. We only examine the other three properties for the uniformity metric $-\mathcal{L}_{\mathcal{U}}$.

First, we prove that $-\mathcal{L}_{\mathcal{U}}$ does not satisfy Property 2. Due to the definition of $\mathcal{L}_{\mathcal{U}}$ in Eqn. (2), we have

$$
\begin{aligned}
\mathcal{L}_{\mathcal{U}}(\mathcal{D} \uplus \mathcal{D}) &:= \log \frac{1}{2n(2n-1)/2} \left( 4 \sum_{i=2}^{n} \sum_{j=1}^{i-1} e^{-t\|\mathbf{z}_i - \mathbf{z}_j\|_2^2} + \sum_{i=1}^{n} e^{-t\|\mathbf{z}_i - \mathbf{z}_i\|_2^2} \right) \\
&= \log \frac{1}{2n(2n-1)/2} \left( 4 \sum_{i=2}^{n} \sum_{j=1}^{i-1} e^{-t\|\mathbf{z}_i - \mathbf{z}_j\|_2^2} + n \right).
\end{aligned}
$$

(10)

Letting $G = \sum_{i=2}^{n} \sum_{j=1}^{i-1} e^{-t\|\mathbf{z}_i - \mathbf{z}_j\|_2^2}$, we have

$$
G = \sum_{i=2}^{n} \sum_{j=1}^{i-1} e^{-t\|\mathbf{z}_i - \mathbf{z}_j\|_2^2} \leq \sum_{i=2}^{n} \sum_{j=1}^{i-1} e^{-t\|\mathbf{z}_i - \mathbf{z}_i\|_2^2} = n(n-1)/2,
$$

and $G = n(n-1)/2$ if and only if $\mathbf{z}_1 = \mathbf{z}_2 = \ldots = \mathbf{z}_n$. Thus

$$\mathcal{L}_{\mathcal{U}}(\mathcal{D} \uplus \mathcal{D}) - \mathcal{L}_{\mathcal{U}}(\mathcal{D}) = \log \frac{4G + n}{2n(2n-1)/2} - \log \frac{G}{n(n-1)/2}$$

$$= \log \frac{(4G+n)n(n-1)/2}{2nG(2n-1)/2} = \log \frac{(4G+n)(n-1)}{4nG - 2G}$$

$$= \log \frac{4nG - 4G + n^2 - n}{4nG - 2G} \geq \log 1 = 0.$$

The above equality holds if and only if $G = n(n-1)/2$, which requires $\mathbf{z}_1 = \mathbf{z}_2 = ... = \mathbf{z}_n$, a trivial case when all representations collapse to one constant point. We have excluded this trivial case, and thus $-\mathcal{L}_{\mathcal{U}}(\mathcal{D} \uplus \mathcal{D}) < -\mathcal{L}_{\mathcal{U}}(\mathcal{D})$. Therefore, the uniformity metric $-\mathcal{L}_{\mathcal{U}}$ does not satisfy Property 2.

Second, we prove that $-\mathcal{L}_{\mathcal{U}}$ does not satisfy Property 3. Letting $\widehat{\mathbf{z}}_i = \mathbf{z}_i \oplus \mathbf{z}_i$ and $\widehat{\mathbf{z}}_j = \mathbf{z}_j \oplus \mathbf{z}_j$, we have

$$\mathcal{L}_{\mathcal{U}}(\mathcal{D} \oplus \mathcal{D}) := \log \frac{1}{n(n-1)/2} \sum_{i=2}^{n} \sum_{j=1}^{i-1} e^{-t\|\widehat{\mathbf{z}}_i - \widehat{\mathbf{z}}_j\|_2^2}.$$

By the definitions of $\widehat{\mathbf{z}}_i$ and $\widehat{\mathbf{z}}_j$, we have $\|\widehat{\mathbf{z}}_i\|_2 = \sqrt{2}\|\mathbf{z}_i\|_2$, $\|\widehat{\mathbf{z}}_j\|_2 = \sqrt{2}\|\mathbf{z}_j\|_2$, and $\langle \widehat{\mathbf{z}}_i, \widehat{\mathbf{z}}_j \rangle = 2\langle \mathbf{z}_i, \mathbf{z}_j \rangle$. Thus

$$\|\widehat{\mathbf{z}}_i - \widehat{\mathbf{z}}_j\|_2^2 = 2\|\mathbf{z}_i\|_2^2 + 2\|\mathbf{z}_j\|_2^2 - 4\langle \mathbf{z}_i, \mathbf{z}_j \rangle = 2\|\mathbf{z}_i - \mathbf{z}_j\|_2^2 \geq \|\mathbf{z}_i - \mathbf{z}_j\|_2^2.$$

Therefore, $-\mathcal{L}_{\mathcal{U}}(\mathcal{D} \oplus \mathcal{D}) \geq -\mathcal{L}_{\mathcal{U}}(\mathcal{D})$, indicating that the uniformity metric $-\mathcal{L}_{\mathcal{U}}$ does not satisfy the Property 3.

Third, we prove that the existing metric $-\mathcal{L}_{\mathcal{U}}$ does not satisfy the Property 4. Letting $\widehat{\mathbf{z}}_i = \mathbf{z}_i \oplus \mathbf{0}^k$ and $\widehat{\mathbf{z}}_j = \mathbf{z}_j \oplus \mathbf{0}^k$, we have

$$\mathcal{L}_{\mathcal{U}}(\mathcal{D} \oplus \mathbf{0}^k) := \log \frac{1}{n(n-1)/2} \sum_{i=2}^{n} \sum_{j=1}^{i-1} e^{-t\|\widehat{\mathbf{z}}_i - \widehat{\mathbf{z}}_j\|_2^2}.$$

By the definitions of $\widehat{\mathbf{z}}_i$ and $\widehat{\mathbf{z}}_j$, we have $\|\widehat{\mathbf{z}}_i\|_2 = \|\mathbf{z}_i\|_2$, $\|\widehat{\mathbf{z}}_j\|_2 = \|\mathbf{z}_j\|_2$, $\langle \widehat{\mathbf{z}}_i, \widehat{\mathbf{z}}_j \rangle = \langle \mathbf{z}_i, \mathbf{z}_j \rangle$, and thus

$$\|\widehat{\mathbf{z}}_i - \widehat{\mathbf{z}}_j\|_2^2 = \|\widehat{\mathbf{z}}_i\|_2^2 + \|\widehat{\mathbf{z}}_j\|_2^2 - 2\langle \widehat{\mathbf{z}}_i, \widehat{\mathbf{z}}_j \rangle = \|\mathbf{z}_i\|_2^2 + \|\mathbf{z}_j\|_2^2 - 2\langle \mathbf{z}_i, \mathbf{z}_j \rangle = \|\mathbf{z}_i - \mathbf{z}_j\|_2^2.$$

Therefore, $-\mathcal{L}_{\mathcal{U}}(\mathcal{D} \oplus \mathbf{0}^k) = -\mathcal{L}_{\mathcal{U}}(\mathcal{D})$, indicating that the uniformity metric $-\mathcal{L}_{\mathcal{U}}$ does not satisfy Property 4.

## C.2 Proof of Theorem 3: Examining the four properties for $-\mathcal{W}_2$

Property 1 can be easily verified for $-\mathcal{W}_2$, and thus the proof is skipped. We only examine the rest three properties for the proposed uniformity metric $-\mathcal{W}_2$.

First, we prove that our proposed metric $-\mathcal{W}_2$ satisfies Property 2. Let $\widehat{\boldsymbol{\mu}}$ and $\widehat{\boldsymbol{\Sigma}}$ be defined as above, for $\mathcal{D} \uplus \mathcal{D} = \{\mathbf{z}_1, \mathbf{z}_2, ..., \mathbf{z}_n, \mathbf{z}_1, \mathbf{z}_2, ..., \mathbf{z}_n\}$, the mean and covariance estimators are

$$\widetilde{\boldsymbol{\mu}} = \frac{1}{2n} \sum_{i=1}^{n} 2\mathbf{z}_i = \widehat{\boldsymbol{\mu}}, \quad \widetilde{\boldsymbol{\Sigma}} = \frac{1}{2n} \sum_{i=1}^{n} 2(\mathbf{z}_i - \widetilde{\boldsymbol{\mu}})^T (\mathbf{z}_i - \widetilde{\boldsymbol{\mu}}) = \widehat{\boldsymbol{\Sigma}},$$

which agree with those for $\mathcal{D}$. Then we have

$$\mathcal{W}_2(\mathcal{D} \uplus \mathcal{D}) := \sqrt{\|\widehat{\boldsymbol{\mu}}\|_2^2 + 1 + \text{tr}(\widehat{\boldsymbol{\Sigma}}) - \frac{2}{\sqrt{m}} \text{tr}(\widehat{\boldsymbol{\Sigma}}^{1/2})} = \mathcal{W}_2(\mathcal{D}).$$

Therefore, our proposed metric $-\mathcal{W}_2$ satisfies Property 2.

Second, we prove that $-\mathcal{W}_2$ satisfies Property 3. Let $\widetilde{\mathbf{z}}_i = \mathbf{z}_i \oplus \mathbf{z}_i \in \mathbb{R}^{2m}$. For $\mathcal{D} \oplus \mathcal{D}$, the mean and covariance estimators are:

$$\widetilde{\boldsymbol{\mu}} = \begin{pmatrix} \widehat{\boldsymbol{\mu}} \\ \widehat{\boldsymbol{\mu}} \end{pmatrix}, \quad \widetilde{\boldsymbol{\Sigma}} = \begin{pmatrix} \widehat{\boldsymbol{\Sigma}} & \widehat{\boldsymbol{\Sigma}} \\ \widehat{\boldsymbol{\Sigma}} & \widehat{\boldsymbol{\Sigma}} \end{pmatrix}.$$

We easily have

$$\widetilde{\boldsymbol{\Sigma}}^{1/2} = \begin{pmatrix} \widehat{\boldsymbol{\Sigma}}^{1/2}/\sqrt{2} & \widehat{\boldsymbol{\Sigma}}^{1/2}/\sqrt{2} \\ \widehat{\boldsymbol{\Sigma}}^{1/2}/\sqrt{2} & \widehat{\boldsymbol{\Sigma}}^{1/2}/\sqrt{2} \end{pmatrix}, \ \ \mathrm{tr}(\widetilde{\boldsymbol{\Sigma}}) = 2\,\mathrm{tr}(\widehat{\boldsymbol{\Sigma}}), \ \text{and} \ \mathrm{tr}(\widetilde{\boldsymbol{\Sigma}}^{1/2}) = \sqrt{2}\,\mathrm{tr}(\widehat{\boldsymbol{\Sigma}}^{1/2}).$$

Thus

$$\begin{aligned}
\mathcal{W}_2(\mathcal{D} \oplus \mathcal{D}) &:= \sqrt{\|\widetilde{\boldsymbol{\mu}}\|_2^2 + 1 + \mathrm{tr}(\widetilde{\boldsymbol{\Sigma}}) - \frac{2}{\sqrt{2m}}\mathrm{tr}(\widetilde{\boldsymbol{\Sigma}}^{1/2})} \\
&= \sqrt{2\|\widehat{\boldsymbol{\mu}}\|_2^2 + 1 + 2\,\mathrm{tr}(\widehat{\boldsymbol{\Sigma}}) - \frac{2\sqrt{2}}{\sqrt{2m}}\mathrm{tr}(\widehat{\boldsymbol{\Sigma}}^{1/2})} \\
&> \sqrt{\|\widehat{\boldsymbol{\mu}}\|_2^2 + 1 + \mathrm{tr}(\widehat{\boldsymbol{\Sigma}}) - \frac{2}{\sqrt{m}}\mathrm{tr}(\widehat{\boldsymbol{\Sigma}}^{1/2})} = \mathcal{W}_2(\mathcal{D}).
\end{aligned}$$

Therefore, $-\mathcal{W}_2(\mathcal{D} \oplus \mathcal{D}) < -\mathcal{W}_2(\mathcal{D})$, indicating that our proposed metric $-\mathcal{W}_2$ could satisfy the Property 3.

Third, we prove that our proposed metric $-\mathcal{W}_2$ satisfies Property 4. Let $\widetilde{\mathbf{z}}_i = \mathbf{z}_i \oplus \mathbf{0}^k \in \mathbb{R}^{m+k}$ with an overload of notation. For $\mathcal{D} \oplus \mathbf{0}^k$, the sample mean and covariance estimators are

$$\widetilde{\boldsymbol{\mu}} = \begin{pmatrix} \widehat{\boldsymbol{\mu}} \\ \mathbf{0}^k \end{pmatrix}, \quad \widetilde{\boldsymbol{\Sigma}} = \begin{pmatrix} \widehat{\boldsymbol{\Sigma}} & \mathbf{0}^{m\times k} \\ \mathbf{0}^{k\times m} & \mathbf{0}^{k\times k} \end{pmatrix},$$

where $\widehat{\boldsymbol{\mu}}$ and $\widehat{\boldsymbol{\Sigma}}$ are defined previously. Therefore, we have $\mathrm{tr}(\widetilde{\boldsymbol{\Sigma}}) = \mathrm{tr}(\widehat{\boldsymbol{\Sigma}})$, $\mathrm{tr}(\widetilde{\boldsymbol{\Sigma}}^{1/2}) = \mathrm{tr}(\widehat{\boldsymbol{\Sigma}}^{1/2})$, and thus

$$\begin{aligned}
\mathcal{W}_2(\mathcal{D} \oplus \mathbf{0}^k) &:= \sqrt{\|\widetilde{\boldsymbol{\mu}}\|_2^2 + 1 + \mathrm{tr}(\widetilde{\boldsymbol{\Sigma}}) - \frac{2}{\sqrt{m+k}}\mathrm{tr}(\widetilde{\boldsymbol{\Sigma}}^{1/2})} \\
&= \sqrt{\|\widehat{\boldsymbol{\mu}}\|_2^2 + 1 + \mathrm{tr}(\widehat{\boldsymbol{\Sigma}}) - \frac{2}{\sqrt{m+k}}\mathrm{tr}(\widehat{\boldsymbol{\Sigma}}^{1/2})} \\
&> \sqrt{\|\widehat{\boldsymbol{\mu}}\|_2^2 + 1 + \mathrm{tr}(\widehat{\boldsymbol{\Sigma}}) - \frac{2}{\sqrt{m}}\mathrm{tr}(\widehat{\boldsymbol{\Sigma}}^{1/2})} = \mathcal{W}_2(\mathcal{D}).
\end{aligned}$$

Therefore, $-\mathcal{W}_2(\mathcal{D} \oplus \mathbf{0}^k) < -\mathcal{W}_2(\mathcal{D})$, indicating that our proposed metric $-\mathcal{W}_2$ satisfies the Property 4.

# D  FURTHER COMPARISONS BETWEEN $\mathbf{Y}$ AND $\widehat{\mathbf{Y}}$

This section further compares the distributions of $\mathbf{Y}$ and $\widehat{\mathbf{Y}}$.

We visually compare the distributions of $Y_i$ and $\widehat{Y}_i$. To estimate the distributions of $Y_i$ and $\widehat{Y}_i$, we bin 200,000 sampled data points into 51 groups. Figure 8 compares the binning densities of $Y_i$ and $\widehat{Y}_i$ when $m \in \{2, 4, 8, 16, 32, 64, 128, 256\}$. We can observe that two distributions are highly overlapped when $m$ is moderately large, e.g., $m \geq 8$ or $m \geq 16$.

By binning 2,000,000 data points into $51 \times 51$ groups in two-axis, we also analyze the joint binning densities and present 2D joint binning densities of $(Y_i, Y_j)$ $(i \neq j)$ in Figure 9(a) and $(\widehat{Y}_i, \widehat{Y}_j)$ $(i \neq j)$ in Figure 9(b). Even if $m$ is relatively small (i.e., 32), the densities of the two distributions are close.

# E  ADDITIONAL SYNTHETIC STUDIES

## E.1  CORRELATION BETWEEN $-\mathcal{L}_{\mathcal{U}}$ AND $-\mathcal{W}_2$

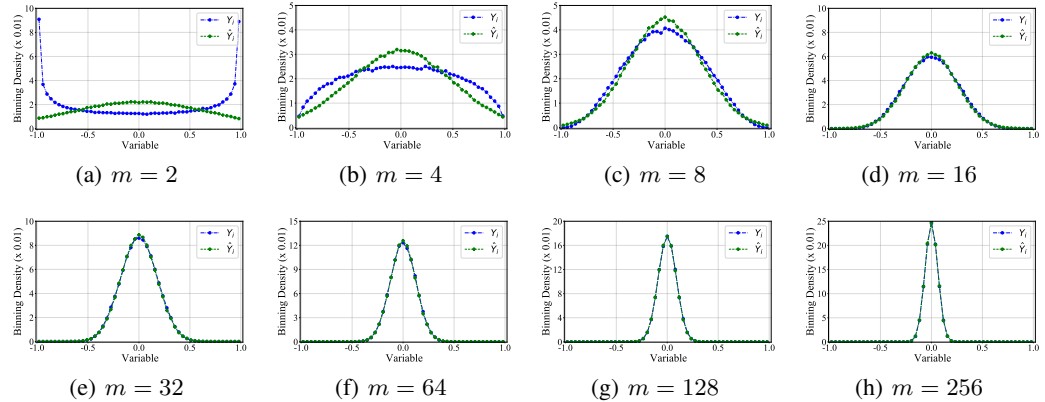

Figure 8: Comparing the binning densities of $Y_i$ and $\widehat{Y}_i$ with various dimensions.

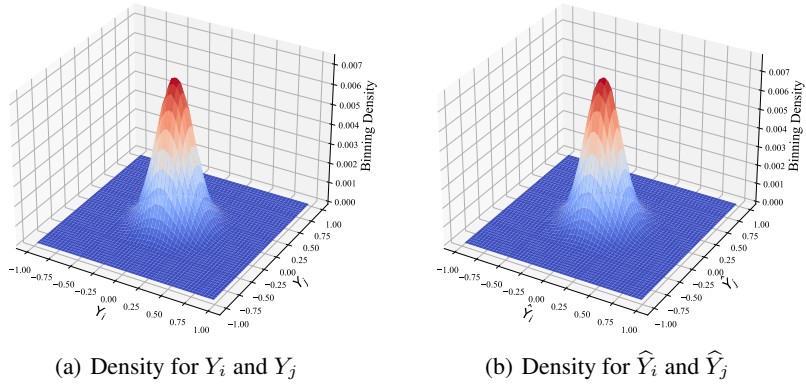

(a) Density for $Y_i$ and $Y_j$      (b) Density for $\widehat{Y}_i$ and $\widehat{Y}_j$

Figure 9: Visualization of two arbitrary dimensions for $\mathbf{Y}$ and $\widehat{\mathbf{Y}}$ when $m = 32$.

We employ synthetic experiments to study the uniformity metrics across different distributions. Specifically, we sample 50,000 data vectors ($m = 256$) from different distributions, such as the isotropic Gaussian distribution $\mathcal{N}(\mathbf{0}, \mathbf{I})$, the uniform distribution on the hyperrectangle $[\mathbf{0}, \mathbf{1}]$, and the mixture of Gaussians, etc. Then we normalize these data vectors, and estimate the uniformity of different distributions by two metrics. As shown in Fig. 10, isotropic Gaussian distribution achieves the maximum values for both $-\mathcal{W}_2$ and $-\mathcal{L}_{\mathcal{U}}$, which indicates that isotropic Gaussian distribution achieves larger uniformity than other distributions. This empirical result is consistent with Fact 1 that the isotropic Gaussian distribution (approximately) achieves the maximum uniformity.

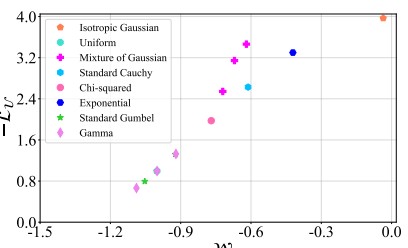

Figure 10: Uniformity analysis for various distributions by two metrics.

### E.2 ON INSTANCE CLONING CONSTRAINT

In this section, we compare the two metrics in terms of Property 2 (ICC). Specifically, we randomly sample 1,000 data vectors from the isotropic Gaussian distribution ($m = 32$) and then mask $50\%$ of their coordinates with zeros, forming a new dataset $\mathcal{D}$ with an overload of notation. To investigate the impact of instance cloning, we create multiple clones of the dataset, such as $\mathcal{D} \uplus \mathcal{D}$ and $\mathcal{D} \uplus \mathcal{D} \uplus \mathcal{D}$, which correspond to one and two times cloning, respectively. We evaluate the two metrics on these datasets. Figure 11 shows that the value of $-\mathcal{L}_{\mathcal{U}}$ slightly decreases as the number of clones increases, in-

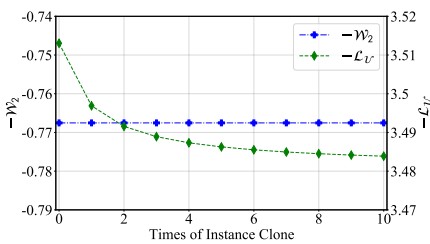

Figure 11: ICC analysis.

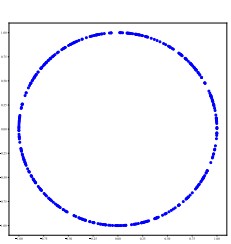 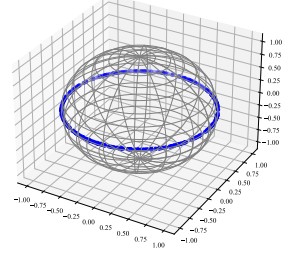 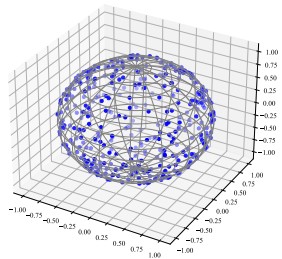

(a) Two-dimensional visu-
alization with no collapsed
dimension

(b) Three-dimensional visualiza-
tion with one collapsed dimen-
sion

(c) Three-dimensional visualiza-
tion with no collapsed dimension

Figure 12: A case study for Property 4 and blue points are data vectors.

dicating that $-\mathcal{L}_\mathcal{U}$ violates the equality in Equation 4. In contrast, our proposed metric $-\mathcal{W}_2$ remains constant, satisfying the equality.

### E.3 UNDERSTANDING PROPERTY 4: WHY DOES IT RELATE TO DIMENSIONAL COLLAPSE?

This section delves into Property 4 through case studies. Let us begin with a thought experiment. Consider a dataset $\mathcal{D}$ with instances uniformly distributed on the unit hypersphere, thereby possessing (almost) maximal uniformity. When additional coordinates with zeros are inserted to each instance of $\mathcal{D}$, forming a new dataset $\mathcal{D} \oplus \mathbf{0}^k$, it can no longer maintain maximal uniformity. This is because, the new dataset only occupies a small area of the unit hypersphere. Consequently, as $k$ increases, the uniformity of the dataset would decrease significantly.

Let us visualize this thought experiment using synthetic studies. In Figure 12(a), we present 400 data vectors ($\mathcal{D}_1$) sampled from $\mathcal{N}(\mathbf{0}, \mathbf{I}_2)$, which are also nearly uniformly distributed on $\mathcal{S}^1$. By inserting one zero-coordinate to each instance of $\mathcal{D}_1$, we obtain a new dataset $\mathcal{D}_1 \oplus \mathbf{0}^1$, as depicted in Figure 12(b). We also construct another dataset $\mathcal{D}_2$ consisting of 400 data vectors sampled from $\mathcal{N}(\mathbf{0}, \mathbf{I}_3)$, visualized in Figure 12(c). Notably, $\mathcal{D}_1 \oplus \mathbf{0}^1$ forms a ring on $\mathcal{S}^2$, while $\mathcal{D}_2$ is almost uniformly distributed over $\mathcal{S}^2$. Naturally, $\mathcal{U}(\mathcal{D}_2) > \mathcal{U}(\mathcal{D}_1 \oplus \mathbf{0}^1)$. If $\mathcal{U}(\mathcal{D}_1) = \mathcal{U}(\mathcal{D}_2)^4$, then $\mathcal{U}(\mathcal{D}_1) = \mathcal{U}(\mathcal{D}_2) > \mathcal{U}(\mathcal{D}_1 \oplus \mathbf{0}^1)$. This partially confirms the validity of Property 4.

Additionally, increasing the value of $k$ in Property 4 exacerbates the degree of dimensional collapse. To illustrate, consider a dataset $\mathcal{D}$ sampled from a multivariate Gaussian distribution $\mathcal{N}(\mathbf{0}, \mathbf{I}_m/m)$, exhibiting a collapse degree close to $0\%$. However, upon inserting $m$-dimensional zero-value vectors to each instance of $\mathcal{D}$, denoted as $\mathcal{D} \oplus \mathbf{0}^m$, half of the dimensions collapse. Consequently, the collapse degree increases to $50\%$. Figure 13 visually represents the collapse of $\mathcal{D} \oplus \mathbf{0}^k$ using the singular value spectra of the representations. It is evident that a larger $k$ results in a more pronounced dimensional collapse. In summary, Property 4 corresponds to dimensional collapse.

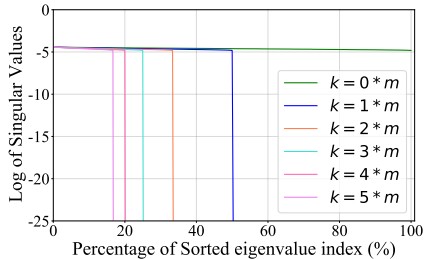

Figure 13: Singular value spectrum of $\mathcal{D} \oplus \mathbf{0}^k$

### E.4 UNDERSTANDING $\mathcal{W}_2$: LARGE MEANS MAY LEAD TO COLLAPSE

In this section, we explore our uniformity loss $\mathcal{W}_2$. This loss embodies two primary constraints. Firstly, it promotes the covariance matrix to be isotropic (specifically $\mathbf{I}_m/m$). Secondly, it enforces the mean to be zero. The latter constraint on the mean is crucial. To illustrate, we present a case study demonstrating that deviating the mean from zero compromises uniformity, even if the covariance matrix is precisely $\mathbf{I}_m/m$ and thus isotropic. Means deviating from zero may result in dimensional collapse and even constant collapse.

---

[4]Intuitively, maximal uniformity should stay constant regardless of dimensions; otherwise the corresponding uniformity metric exhibit a preference for larger or smaller dimensions.

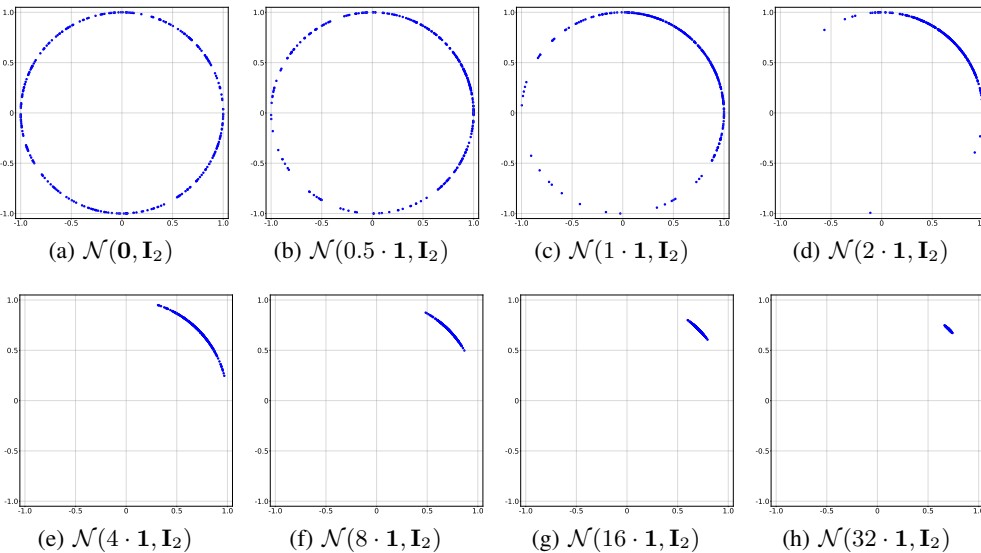

Figure 14: Visualizing $\ell_2$ normalized Gaussian vectors with different means.

Table 3: Parameter settings for various models in the experiments.

| Models | MoCo v2 | BYOL | BarlowTwins | Zero-CL |
|---|---|---|---|---|
| $\alpha_{\max}$ | 1.0 | 0.2 | 30.0 | 30.0 |
| $\alpha_{\min}$ | 1.0 | 0.2 | 0 | 30.0 |

Assuming $\mathbf{X} \in \mathbb{R}^2$ follows a Gaussian distribution $\mathcal{N}(\mathbf{0}, \mathbf{I}_2)$, let $\mathbf{Y} = \mathbf{X} + k \cdot \mathbf{1}$ such that $\mathbf{Y} \sim \mathcal{N}(k \cdot \mathbf{1}, \mathbf{I}_2)$, where $\mathbf{1} \in \mathbb{R}^k$ represents a vector of all ones. We vary $k$ from 0 to 32 and visualize the $\ell_2$-normalized $\mathbf{Y}$'s in Figure 14 (by generating multiple independent copies). It is clear that an excessively large means will cause representations to collapse to a single point, even if the covariance matrix is isotropic.

## F    EXPERIMENT SETTINGS AND CONVERGENCE ANALYSIS

### F.1    EXPERIMENT SETTINGS

To ensure fair comparisons, all experiments in Section 6 are conducted on a single 1080 GPU. Additionally, we maintain consistency in network architecture across all models, utilizing ResNet-18 (He et al., 2016) as the backbone and a three-layer MLP as the projector. The LARS optimizer (You et al., 2017) is employed with a base learning rate of $0.2$, accompanied by a cosine decay learning rate schedule (Loshchilov & Hutter, 2017) for all models. Evaluation follows a linear evaluation protocol, where models are pre-trained for 500 epochs. Evaluation involves adding a linear classifier and training the classifier for 100 epochs while preserving the learned representations. The same augmentation strategy is deployed across all models, encompassing various operations such as color distortion, rotation, and cutout. Following da Costa et al. (2022), we set the temperature $t = 0.2$ for all contrastive learning methods. For MoCo (He et al., 2020) and NNCLR (Dwibedi et al., 2021), which require an additional queue to store negative samples, we set the queue size to $2^{12}$. Regarding the linear decay for weighting the quadratic Wasserstein distance, refer to Table 3 for the parameter settings.

### F.2    CONVERGENCE ANALYSIS FOR TOP-1 ACCURACY

Here we illustrate the convergence of Top-1 accuracy across all training epochs in Fig 15. Throughout the training, we capture the model checkpoint at the end of each epoch to train a linear classifier. We subsequently evaluate the Top-1 accuracy on unseen images from the test set (either CIFAR-10 or CIFAR-100).

For both CIFAR-10 and CIFAR-100, we observe that integrating the proposed uniformity metric as an auxiliary loss significantly enhances the Top-1 accuracy, particularly in the initial stages of training.

### F.3 CONVERGENCE ANALYSIS FOR UNIFORMITY AND ALIGNMENT

This section presents the convergence of the uniformity metric and alignment loss across all training epochs in Figure 16 and Figure 17, respectively. Throughout the training, we record the model checkpoint at the end of each epoch to evaluate the uniformity using the proposed metric $\mathcal{W}_2$ and alignment (Wang & Isola, 2020) on unseen images from the test set (either CIFAR-10 or CIFAR-100).

For both CIFAR-10 and CIFAR-100, we observe that integrating the proposed uniformity metric as an auxiliary loss significantly improves uniformity. However, it also slightly compromises alignment (where a smaller alignment loss indicates better alignment). It should be noted that improved uniformity often leads to worse alignment.

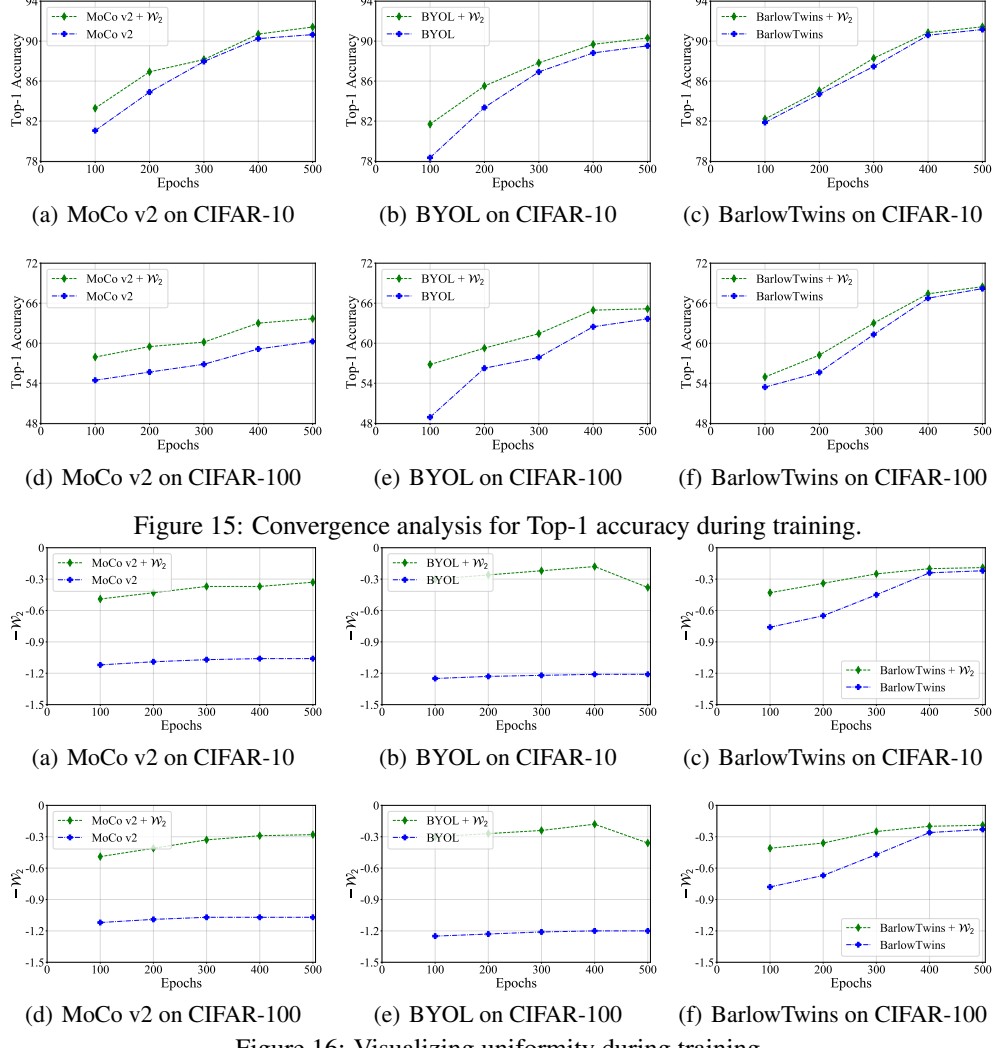

Figure 15: Convergence analysis for Top-1 accuracy during training.

Figure 16: Visualizing uniformity during training

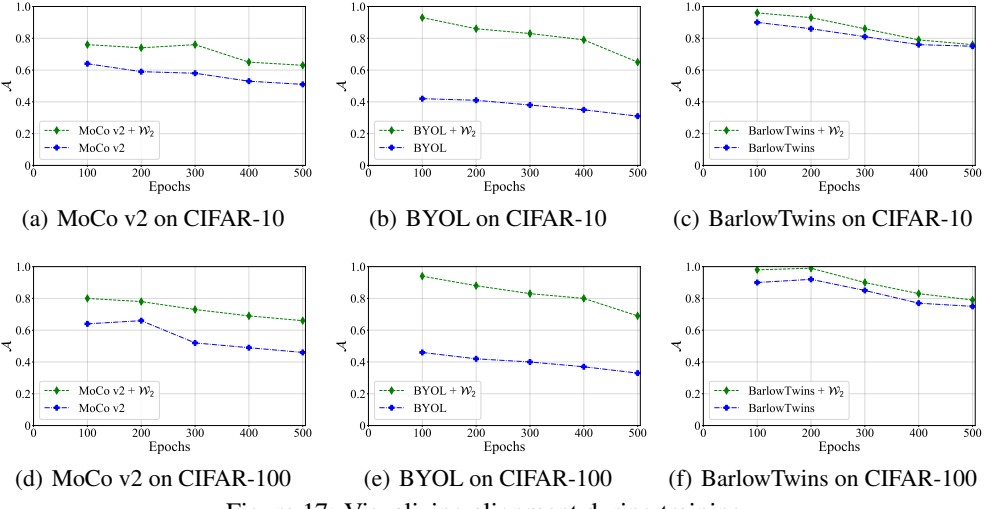

(a) MoCo v2 on CIFAR-10     (b) BYOL on CIFAR-10     (c) BarlowTwins on CIFAR-10

(d) MoCo v2 on CIFAR-100     (e) BYOL on CIFAR-100     (f) BarlowTwins on CIFAR-100

Figure 17: Visualizing alignment during training.

