# OpenReview forum: "Rethinking the Uniformity Metric in Self-Supervised Learning"
_ICLR.cc/2024/Conference — ICLR 2024 poster_

### Official Review · Reviewer_u9aL · 2023-10-27

**Soundness:** 3 good
**Presentation:** 3 good
**Contribution:** 2 fair
**Rating:** 5
**Confidence:** 4

**Summary:**

This study develops a new metric assessing the robustness of learnt representations by measuring collapse degree of them. The authors first point out the weakness of previous metrics proposed by Wang & Isola (2020) in its insensitivity with dimensional collapse. In addition, five characteristics are determined as the criteria for the ideal metric. The new metric (*Wasserstein Distance* between normalized learnt representations’ distribution with zero-mean isotropic Gaussian distribution) is introduced satisfying all these criteria, and illustrates its sensitivity towards dimensional collapse. Empirically, this metric help boosting existing frameworks’ performance in downstream tasks.


Reference:

Wang, T., & Isola, P. (2020, November). Understanding contrastive representation learning through alignment and uniformity on the hypersphere. In *International Conference on Machine Learning* (pp. 9929-9939).

**Strengths:**

- Contribution:
    - This study stems from a good motivation: The lack of current researches explicitly dealing with representation collapse - here dimensional collapse.
    - 5 listed criteria for the ideal ‘uniformity’ metric are logically/mathematically sensible. By pointing out the failure of previous work’s metric (Wang & Isola, 2020), the authors clarify the need for a new metric.
    - The experiments (e.g. Table 2) compare various vanilla approaches with their variance with the proposed metric. These are representative frameworks following three existing directions in dealing with constant collapse. This suggest the potential robustness of proposed metrics when being incorporated to a wide range of models.

- Presentation:
    - The intuitive flow makes it easy catch on what the authors want to deliver.
    - The authors provide trackable mathematical notations and derivations.

**Weaknesses:**

###

- While the 5 criteria are sensible, they can be considered as **necessary conditions** for a good metric. However, no assessment made to ensure they are **sufficient** to construct an ideal one.
- The key contribution of the work - new ‘uniformity’, currently rely on the assumption that the learnt representations follow a Gaussian distribution, which, in turn, can fall apart and the metric can cause undesired effect.
- For experiments, only what related to collapse analysis there exists the comparison between Wang & Isola’s ‘uniformity’ metric with the proposed one. We can conclude nothing on the better performance of the proposed WD-based loss when fused with existing frameworks. It seems not convincing for developing a new loss if the existing loss still works better.

**Questions:**

- Can the authors empirically justify if the proposed loss outperform existing loss when being incorporated with other frameworks?

---

> ### Author Response · Authors · 2023-11-17
> **Response to reviewer u9aL**
>
> **R 4.1 While the 5 criteria are sensible, they can be considered as **necessary conditions** for a good metric. However, no assessment made to ensure they are **sufficient** to construct an ideal one.**
>
> We have pointed it out as one limition of our work in the conclusion. We emphatically recognize and agree that our identified five attributes are necessary, though they may not be entirely sufficient. For example, we could introduce a concept called 'Feature Unactivation Constraint'. This suggests that using zero values to mask parts of the feature dimension could aggravate dimensional collapse. Unlike the existing Feature Baby Constraint (Property 5), the feature unactivation constraint operates under the assumption that the dimensions of the two data sets are the same. However, we felt difficult to theorecially validate the properties for the existing metric and the proposed metrics, as in Claim 1 and Appendix C.1.
>
> Our goal, however, extends beyond defining numerous properties; we aim to identify core, atomic attributes. The overarching objective is to create a framework where these atomic attributes could possibly give rise to a more extensive set of properties, all of which are capable of being theoretically validated.
>
> **R 4.2 The key contribution of the work - new ‘uniformity’, currently rely on the assumption that the learnt representations follow a Gaussian distribution, which, in turn, can fall apart and the metric can cause undesired effect.**
>
> We acknowledge that relying on the assumption of a Gaussian distribution may have undesired effects. However, it is worth noting that this assumption is commonly employed in various practices. For instance, the Fréchet inception distance (FID), a widely used metric for evaluating generated images from generative models like diffusion models, assumes that both the generated images and the ground-truth images follow a Gaussian distribution. Despite its limitations, FID remains a popular choice for measuring the distance between images in different domains.
>
> It is possible that the Law of Large Numbers plays a role in ensuring that learned representations tend to follow a Gaussian distribution to some extent. This statistical principle suggests that as the sample size increases, the sample mean converges to the population mean. Therefore, with a sufficiently large dataset, the learned representations may exhibit Gaussian-like characteristics.

---

> ### Author Response · Authors · 2023-11-17
> **Response to reviewer u9aL (part2)**
>
> **R 4.3 Can the authors empirically justify if the proposed loss outperform existing loss when being incorporated with other frameworks?**
>
> We thank the reviewer for this question. In the revised version, we have added a group of experiments that  incorporates  the existing uniformity loss  (showed as **+ $\mathcal{L_U}$**) into the MoCo v2, BYOL, BarlowTwins, and Zero-CL frameworks. As shown in Table 2, our proposed loss, denoted as $\mathcal{W}_{2}$, outperforms the existing loss in terms of Top-1 accuracy. Additionally, Figure 7(b) and Figure 7(c) demonstrate that $\mathcal{L_U}$ has a minimal effect in preventing dimensional collapse.
>
> | Methods | Proj. | Pred. | CIFAR-10 Acc@1↑ | CIFAR-10 Acc@5↑ | CIFAR-10 $\mathcal{W}_{2}$↓ | CIFAR-10 $\mathcal{L_U}$↓ | CIFAR-10 $\mathcal{A}$↓ | CIFAR-100 Acc@1↑ | CIFAR-100 Acc@5↑ | CIFAR-100 $\mathcal{W}_{2}$↓ | CIFAR-100 $\mathcal{L_U}$↓ | CIFAR-100 $\mathcal{A}$↓ |
> | --- | --- | --- | --- | --- | --- | --- | --- | --- | --- | --- | --- | --- |
> | MoCo v2 | 256 | ✘ | 90.65 | 99.81 | 1.06 | -3.75 | 0.51 | 60.27 | 86.29 | 1.07 | -3.60 | 0.46 |
> | MoCo v2 + $\mathcal{L_U}$ | 256 | ✘ | 90.98 ↑₀.₃₃ | 99.67 | 0.98 ↑₀.₀₈ | -3.82 | 0.53 ↓₀.₀₂ | 61.21 ↑₀.₉₄ | 87.32 | 0.98 ↑₀.₀₉ | -3.81 | 0.52 ↓₀.₀₆ |
> | MoCo v2 + $\mathcal{W}_{2}$ | 256 | ✘ | 91.41 ↑₀.₇₆ | 99.68 | 0.33 ↑₀.₇₃ | -3.84 | 0.63 ↓₀.₁₂ | 63.68 ↑₃.₄₁ | 88.48 | 0.28 ↑₀.₇₉ | -3.86 | 0.66 ↓₀.₂₀ |
> | BYOL | 256 | 256 | 89.53 | 99.71 | 1.21 | -2.99 | 0.31 | 63.66 | 88.81 | 1.20 | -2.87 | 0.33 |
> | BYOL + $\mathcal{L_U}$ | 256 | ✘ | 90.09 ↑₀.₅₆ | 99.75 | 1.09 ↑₀.₁₂ | -3.66 | 0.40 ↓₀.₀₉ | 62.68 ↓₀.₉₈ | 88.44 | 1.08 ↑₀.₁₂ | -3.70 | 0.51 ↓₀.₁₈ |
> | BYOL + $\mathcal{W}_{2}$ | 256 | 256 | 90.31 ↑₀.₇₈ | 99.77 | 0.38 ↑₀.₈₃ | -3.90 | 0.65 ↓₀.₃₄ | 65.16 ↑₁.₅₀ | 89.25 | 0.36 ↑₀.₈₄ | -3.91 | 0.69 ↓₀.₃₆ |
> | BarlowTwins | 256 | ✘ | 91.16 | 99.80 | 0.22 | -3.91 | 0.75 | 68.19 | 90.64 | 0.23 | -3.91 | 0.75 |
> | BarlowTwins + $\mathcal{L_U}$ | 256 | ✘ | 91.38 ↑₀.₂₂ | 99.77 | 0.21 ↑₀.₀₁ | -3.92 | 0.76 ↓₀.₀₁ | 68.41 ↑₀.₂₂ | 90.99 | 0.22 ↑₀.₀₁ | -3.91 | 0.76 ↓₀.₀₁ |
> | BarlowTwins + $\mathcal{W}_{2}$ | 256 | ✘ | 91.43 ↑₀.₂₇ | 99.78 | 0.19 ↑₀.₀₃ | -3.92 | 0.76 ↓₀.₀₁ | 68.47 ↑₀.₂₈ | 90.64 | 0.19 ↑₀.₀₄ | -3.91 | 0.79 ↓₀.₀₄ |
> | Zero-CL | 256 | ✘ | 91.35 | 99.74 | 0.15 | -3.94 | 0.70 | 68.50 | 90.97 | 0.15 | -3.93 | 0.75 |
> | Zero-CL + $\mathcal{L_U}$ | 256 | ✘ | 91.28 ↓₀.₀₇ | 99.74 | 0.15 | -3.94 | 0.72 ↓₀.₀₂ | 68.44 ↓₀.₀₆ | 90.91 | 0.15 | -3.93 | 0.74 ↑₀.₀₁ |
> | Zero-CL + $\mathcal{W}_{2}$ | 256 | ✘ | 91.42 ↑₀.₀₇ | 99.82 | 0.14 ↑₀.₀₁ | -3.94 | 0.71 ↓₀.₀₁ | 68.55 ↑₀.₀₅ | 91.02 | 0.14 ↑₀.₀₁ | -3.94 | 0.76 ↓₀.₀₁ |
>
> To further emphasize the effectiveness of our proposed metric in evaluating the uniformity of learned representations, we conducted a case study. Table 2 presents the results for MoCo v2, MoCo v2 + $\mathcal{L_U}$, and MoCo v2 + $\mathcal{W}_{2}$ models on the CIFAR-100 dataset. The corresponding values of $\mathcal{L_U}$ are -3.60, -3.81, and -3.86, respectively. However, these quantitative results alone cannot fully distinguish the degrees of dimensional collapse in Figure 7 (b). In contrast, our proposed metric provides values of 1.07, 0.98, and 0.28 for these models, which are more distinctive and align well with the qualitative analysis.

---

> > ### Comment · Reviewer_u9aL · 2023-11-21
> > **Additional Review from Reviewer u9aL**
> >
> > Dear Authors,
> >
> > Thank you for the effort in addressing my comments as well as modifying the manuscript.
> > The 5 characteristics are quite intuitive and easily verified. I agree they are needed and can be served as core, atomic attributes for further identifications of other properties.
> >
> > However, I still have some additional concern on top of your comments:
> > - *This statistical principle suggests that as the sample size increases, the sample mean converges to the population mean. Therefore, with a sufficiently large dataset, the learned representations may exhibit Gaussian-like characteristics.*
> >
> > I do not think your later claim can be inferred from your first claim, the theorem itself is not really related to the shape of sample distribution.
> >
> > - *our proposed loss outperforms the existing loss in terms of Top-1 accuracy*
> >
> > With these experiment, I can only see the marginal improvement of proposed loss. To justify for the novelty, I suggest the authors verify more extensively with other tasks or datasets.
> >
> > With these concerns, I would keep my current score for now.
> >
> > Best,

---

> ### Author Response · Authors · 2023-11-21
> **further responses**
>
> Thank you for your quick replies. We would like to follow up with your comments.
>
> 1. We'd like to apologize for our confusing sentences (we will revise that). This is largely due to a language writing issue. We meant that due to the central limit theorem a rescaled summation approaches to the Gaussian distribution, thus we use Gaussian distribution as a simplification. Indeed, the proof of Theorem 2 heavily relies on the Gaussian distribution assumption. This is because the calculation of the KL divergence and the Wasserstein distance requires the explicit distribution. Additionally, when converting the upper bound for KL divergence to the upper bound for the quadratic Wasserstein distance, we need the $T_2$ inequality, which also requires the Gaussian assumption. While a different distribution, i.e., the log-concave distribution, might be possible, we do not see the value of extending to such a distribution.
>
> 2. We'd like to first point out that our proposed metric consistently outperforms the method without any alignment metric and the metric by Wang and Isola 2020.  In the case of CIFAR 100, using our metric improves the BYOL without any alignment metric and with Wang and Isola's metric by 1.50 and 0.98 respectively. Similarly, using our metric improves the MOCO v2 without any alignment metric and with Wang and Isola's metric by 3.41 and 0.94 respectively. We consider these as significant improvements. Due to time constraints, we have not done other downstream tasks, but we will also verify our new uniformity metric using other downstream tasks.
>
> Let us know if you have any other questions or concerns, and we hope to interact with you further to improve the quality of the paper. We again would like to thank you for your time and effort in reviewing our paper.

---

### Official Review · Reviewer_a1G3 · 2023-10-29

**Soundness:** 2 fair
**Presentation:** 2 fair
**Contribution:** 2 fair
**Rating:** 5
**Confidence:** 5

**Summary:**

In this paper, the author revisits the alignment and uniformity property in Self-Supervised Learning and shows that the metric in [1] lacks the ability to measure dimensional collapse, which is a phenomenon where the representations learned through self-supervised learning span a low-dimensional subspace instead of being distributed uniformly in the representation space. Therefore, the author proposes a new metric utilizing the Wasserstein distance between the distribution of the representation space and the normalized isotropic Gaussian distribution.

[1] Tongzhou Wang and Phillip Isola. Understanding contrastive representation learning through alignment and uniformity on the hypersphere. In ICML, 2020.

**Strengths:**

1. The paper includes a lot of empirical analysis to demonstrate the effectiveness of the proposed metric.
2. The paper is well-written and easy to follow.

**Weaknesses:**

1. The font size of the figures is too small.
2. It should be made more clear why the original metric doesn't correspond to the proposed 5 properties and how these properties are related to dimensional collapse.
3. It should be made more clear why dimensional collapse is a undesired property in self-supervised learning.
4. Since $\mathcal{W}_2$ is also related to the covariance matrix of representation, what is the relationship between the proposed method and Barlow Twins/VICReg?
5. Why is the KL divergence of $Y$ and $\hat{Y}$ not presented for comparison as in Figure 3?
6. As in [2], the phenomenon of dimensional collapse happens in the embedding space. However, in [2], when the projector is presented, the representation space doesn't collapse. Therefore, why do the singular values of the representation in Figure 8 collapse?
[2] Li Jing, Pascal Vincent, Yann LeCun, and Yuandong Tian. Understanding dimensional collapse in contrastive self-supervised learning. In ICLR, 2022.

**Questions:**

Please refer to Weaknesses

---

> ### Author Response · Authors · 2023-11-17
> **Response to reviewer a1G3**
>
> **R 3.1 It should be made more clear why the original metric doesn't correspond to the proposed 5 properties and how these properties are related to dimensional collapse.**
>
> Thanks for your comment. To explain why the original metric  $-\mathcal{L_U}$ fails to meet proposed constraints, we have provided both theoretical evidence and empirical validations in the revised version.
>
> **Theoretical analysis** In Section 5.1, we have provided  theoretical analysis on the two metrics w.r.t  the five desiderata constraints, as shown in Table 1. Table 1 indicates that the original metric cannot meet ICC, FCC, and FBC. Please see detailed proof in Appendix C.2.
>
> **Experimental comparison** In Section 5.2 of the revised manuscript, we have provided empirical comparisons between two metrics in terms of ICC, FCC, and FBC. As visualized in Figures 4,5,6, the original metric cannot satisfy these constraints.
>
> In summary, our theoretical analysis and empirical results arrive at the same conclusion: the original metric does not satisfy ICC, FCC, and FBC.
>
> **FBC and dimensional collapse** We further explain why FBC is highly related to dimensional collapse. Increasing the value of $k$ in Property 5 would exacerbate the degree of dimensional collapse. To illustrate this, let’s consider a set of data vectors ($\mathcal{D}$) as defined in Section 3.1, sampled from an isotropic Gaussian  distribution, $\mathcal{N}(0, I_m)$. These data vectors are uniformly distributed on the unit hypersphere, resulting in a collapse degree of 0%. However, when we append $m$ dimension zero-value vectors to each data in $\mathcal{D}$, denoted as $\mathcal{D} \oplus  0^{m}$, half of the dimensions become collapsed. As a result, the collapse degree increases to 50%. Furthermore, a larger $k$ would lead to a more severe dimensional collapse. An intuitive visualization can be found in Figure 17 in Appendix P.
>
> **R 3.2 It should be made more clear why dimensional collapse is an undesired property in self-supervised learning.**
>
> In line with Occam's Razor, we should avoid unnecessary multiplication of entities. Generally speaking, an excess of dimensions can lead to redundancy. Although redundancy might be a frequent aspect of modern machine learning and is NOT always harmful- especially in the era of large language models - it is decidedly less efficient in contexts where conserving energy and resources is key. Our preference leans strongly towards the Principle of Parsimony.
>
> **R 3.3 Since $\mathcal{W}_{2}$ is also related to the covariance matrix of representation, what is the relationship between the proposed method and Barlow Twins/VICReg?**
>
> We point out one possible commonality and two differences. The commonality is that both methods aim to decorrelate the covariance matrices and reduce redundancy. This is evident from the fact that the optimal solution in Equation (12) is $\Sigma = \frac{1}{m} I_m$.
>
> Also, there are two differences between the approaches.
>
> - Firstly, the optimization objective for the covariance matrix is slightly different. Barlow Twins/VICReg aims to enforce  $\Sigma = I_m$, while our approach targets $\Sigma = \frac{1}{m} I_m$.
> - Secondly, the motivation behind regularization on the covariance matrix differs. Our proposed $\mathcal{W}_{2}$ loss stems from a distributional perspective, focusing on the distribution of the representations. On the other hand, Barlow Twins/VICReg starts from a redundancy perspective, aiming to reduce redundancy in the representations.
>
> **R 3.4 Why is the KL divergence of $Y$ and $\hat{Y}$ not presented for comparison as in Figure 3.**
>
> Thanks for your suggestions. In the revised version, we apply the $T_2$-inequality in [1] to prove the quadratic Wasserstein distance between $Y_i$ and $\hat{Y}_i$ also goes to zero as $m \to \infty$; please see Theorem 2. As our metric is based on the quadratic Wasserstein metric, it is reasonable to plot the Wasserstein distance instead of the KL divergence. Nevertheless, to respond to the reviewers, we also present the KL divergence of $Y_i$ and $\hat{Y}_i$ in Figure 9(a) of Appendix E.
>
> [1]  Probability in high dimension. Lecture Notes (Princeton University), 2016. (Theorem 4.31)

---

> > ### Comment · Reviewer_a1G3 · 2023-11-22
> > **Response to authors**
> >
> > The authors have addressed some of my concerns. However, I am still care about the significance of dimensional collapse in designing the metric. Also, I think the difference between the proposed method and Barlow Twins/VICReg are minimal. Therefore, I raise my rating, but tend to weak reject.

---

> ### Author Response · Authors · 2023-11-17
> **Response to reviewer a1G3 (part2)**
>
> **R 3.5 the phenomenon of dimensional collapse happens in the embedding space. However, when the projector is presented, the representation space doesn't collapse. Therefore, why do the singular values of the representation in Figure 8 collapse?**
>
> Thanks for your kind comment.  Indeed, the dimensional collapse, as stated in [2], was relieved by using a  project. However, the measured positions for dimensional collapse are different from ours.
>
>
> We denote three types of representations (in three different positions)
>
> - without a projector: $Z_1$
>
> - with  a projector:
>     - the feature before the projector : $Z_2$
>     - the feature after the project:
> $Z_3$
>
> ​
> **Comparison of dimensional collapse between $Z_1$ and $Z_2$**
> In Appendix O, we showed the singular value spectrum for the above three distinct types of representations. Figure 16 illustrates that $Z_2$ has less dimensional collapse compared to $Z_1$.  This is aligned with the conclusion stated in [2].  This finding confirms that adding a projector does mitigate the issue of dimensional collapse.  Note that $Z_1$ still shows a dimensional collapse to some extent.
>
>
>
> **Dimension collapse on $Z_3$**
> Moreover, a crucial observation from Figure 16 is that the representations post-projector, labeled as $Z_3$, are still prone to significant dimensional collapse. This indicates a need for further efforts to reduce dimensional collapse.
>
> Furthermore, the singular value spectra of these post-projector representations are visually represented in Figure 7(a). This depiction makes it apparent that most models, including some recent approaches like SimCLR, BYOL, and MoCo v2, suffer from severe dimensional collapse.
>
> We have clarified the above in the revised version, see  Figure 7(a),  Figure 16, and related discussions.
>
> [2] Understanding dimensional collapse in contrastive self-supervised learning
>
>
>
> **R 3.6 The font size of the figures is too small.**
>
> We have fixed the issue in the revised manuscript.

---

> ### Author Response · Authors · 2023-11-21
> **Rebuttal discussion**
>
> Hello, we are writing again in hopes that you will respond to our reviews and point to specific concerns that remain so that we might have the chance to respond to them.

---

> ### Author Response · Authors · 2023-11-22
> **Rebuttal discussion**
>
> Dear Reviewer a1G3,
>
> We recognize that the timing of this discussion period may not align perfectly with your schedule, yet we would greatly value the opportunity to continue our dialogue before the deadline approaches.
>
> Could you let us know if your questions have been adequately addressed? If not, please feel free to raise them, and we are more than willing to provide further clarification; if you find that your concerns have been resolved, we would appreciate if you could re-consider the review score.
>
> We hope that we have resolved all your questions, but please let us know if there is anything more.
>
> Best wishes to you!

---

> ### Author Response · Authors · 2023-11-22
> **On the  difference between the proposed method and Barlow Twins/VICReg**
>
> Dear Reviewer a1G3,
>
> Thank you for taking the time to review and raising the score.  We are happy to provide further clarifications to address your concerns.
>
>
> As mentioned in R3.3, the optimization objectives and motivations of the two papers are different. Additionally, we want to highlight a significant difference in the working mechanism itself. Unlike Barlow Twins/VICReg, which deals with representational collapse in the **covariance matrix**, we impose additional constraints on the **mean**.
>
> **The constraint on the mean** is not trivial. We present a counterexample to demonstrate that it satisfies the constraints of the covariance matrix, but as the mean changes, its uniformity is affected, leading to a degree of collapse. In extreme cases, it may collapse to a single point.
>
> Assume $ \mathbf{X}$ follows a Gaussian distribution, $ \mathbf{X} \sim \mathcal{N}(0, I_m)$. By adding an additional vector to change its mean, we obtain $ \mathbf{Y}$, where $ \mathbf{Y} =  \mathbf{X} + k \mathbf{I}$ and $ \mathbf{Y} \sim \mathcal{N}(k, I_m)$. $ \mathbf{I}$ is a vector of all ones, and $k$ is a constant.  We could observe that *as $k$ changes, the uniformity of the $\hat{\mathbf{Y}}$ (the $\ell_2$-normalized $ \mathbf{Y}$) will change accordingly*.
>
> See an example code :
> ```python
>     X = torch.randn(N, dim)
>     offset = torch.ones(N, dim)
>
>     X = ( X + mean *offset ).cuda()
>     Y = F.normalize(X, dim=-1)
> ```
>
> The uniformity measured by $-\mathcal{W}_{2}$ is
> | Dim \ Mean  | 0.0  | 0.5  | 1.0  | 2.0  |  4.0  | 8.0  | 16.0 | 32.0 |
> |---------|------|------|------|------|------|------|------|------|
> | 2       | -0.02| -0.43| -0.78| -1.15| -1.31| -1.37| -1.39| -1.40|
> | 4       | -0.01| -0.45| -0.79| -1.11| -1.27| -1.35| -1.38| -1.40|
> | 8       | -0.02 | -0.46| -0.78| -1.08| -1.25| -1.33| -1.37| -1.39|
> | 16      | -0.02 | -0.46 | -0.77| -1.06| -1.24| -1.33| -1.37| -1.39|
> | 32      | -0.03| -0.46 | -0.77| -1.06| -1.24| -1.33| -1.37| -1.39|
> | 64      | -0.04| -0.46 | -0.77| -1.06| -1.23| -1.33| -1.37| -1.39|
>
> In terms of $-\mathcal{L_U}$, it is
>
> | Dim \ Mean | 0.0 | 0.5  | 1.0  | 2.0  | 4.0  | 8.0  | 16.0 | 32.0 |
> |------------|------|------|------|------|------|------|------|------|
> | 2          | 1.55 | 1.30 | 0.87 | 0.36 | 0.11 | 0.03 | 0.01 | 0.00 |
> | 4          | 2.42 | 1.97 | 1.29 | 0.56 | 0.18 | 0.05 | 0.01 | 0.00 |
> | 8          | 3.08 | 2.47 | 1.60 |  0.67 | 0.20 | 0.05 | 0.01 | 0.00 |
> | 16         | 3.51 | 2.81 | 1.78 | 0.74 | 0.22 | 0.06 | 0.01 | 0.00 |
> | 32         | 3.75 | 2.99 | 1.89 | 0.77 | 0.23 | 0.06 | 0.02 | 0.00 |
> | 64         | 3.88 | 3.09 | 1.95 | 0.78 | 0.23 | 0.06 | 0.02 | 0.00 |
>
>
> It could be concluded that
> > Even if the covariance matrix is an identity matrix, an excessively large mean will also cause the representation to collapse.
>
> This shows the mean value affects the uniformity. Considering that the proposed $-\mathcal{W}_{2}$ has an additional constraint on the mean value,  we therefore believe that the proposed metric might be a more principled metric for uniformity.  Refer to Appendix Q and Figure 18 in the revised version for illustrative figures demonstrating the collapse as the mean increases.

---

### Official Review · Reviewer_KkL8 · 2023-10-30

**Soundness:** 3 good
**Presentation:** 2 fair
**Contribution:** 3 good
**Rating:** 6
**Confidence:** 2

**Summary:**

This paper explains the shortcomings of the existing uniformity metric and the potential dimensional collapse then the authors propose new metrics of uniformity. Numerous experiments have proved its effectiveness.

**Strengths:**

The uniformity metrics proposed in this paper are more comprehensive and their validity has been verified experimentally.

**Weaknesses:**

Can the authors further elaborate on the importance of ICC, FCC and FBC to better understand the difference between the proposed metric and previous metrics?

**Questions:**

Why the new metric has a relatively small performance increase on zero-CL?

**Details Of Ethics Concerns:**

No.

---

> ### Author Response · Authors · 2023-11-17
> **Response to Reviewer KkL8**
>
> **R 2.1 Can the authors further elaborate on the importance of ICC, FCC and FBC to better understand the difference between the proposed metric and previous metrics?**
>
> In the revised version, we have added empirical comparisons in Section 5.2 to complement the theoretical comparisons between the two metrics discussed in Section 5.1. The empirical results confirmed that our proposed metric  meets the ICC, FCC, and FBC constraints, while the existing metric fails to do so.
>
> To further explain the constraints of ICC, FCC, and FBC, we provide additional details as suggested by constructive reviewers:
>
> - **ICC**: This constraint ensures that there is no ambiguity in uniformity metrics. We assume that the uniformity of any fixed data distribution is unique, which provides clarity and avoids ambiguity. Since instance cloning does not change the distribution of embeddings, the uniformity of $\mathcal{D} \cup \mathcal{D}$ should be equivalent to that of $\mathcal{D}$. However, the existing metric $-\mathcal{L_U}$ performs an inequality, indicating the ambiguity of the metric.
> - **FCC**: This constraint is aimed at capturing the redundancy in representations caused by feature cloning. When we clone features, half of the dimensions become redundant and essentially useless. Our proposed metric performs an inequality between $\mathcal{D} \oplus \mathcal{D}$ and $\mathcal{D}$, illustrating that it captures this redundancy in learned representations. In contrast, the existing metric performs an equality, failing to capture this redundancy.
> - **FBC**: This constraint is closely related to dimensional collapse. Increasing the value of $k$ in Property 5 would exacerbate the degree of dimensional collapse. To illustrate this, let’s consider a set of data vectors ($\mathcal{D}$) as defined in Section 3.1, sampled from a multivariate normal distribution, $\mathcal{N}(0, I_m)$. These data vectors are uniformly distributed on the unit hypersphere, resulting in a collapse degree of 0%. However, when we insert $m$ dimension zero-value vectors to $\mathcal{D}$, denoted as $\mathcal{D} \oplus  0^{m}$, half of the dimensions become collapsed. As a result, the collapse degree increases to 50%. More detailed explanations and examples can be found in Appendix P.
>
> **R 2.2 Why the new metric has a relatively small performance increase on Zero-CL?**
>
> Zero-CL proposes to whiten representations from both instance-wise and feature-wise perspectives. Specifically, the feature-wise whitening decorrelates representations and can prevent dimensional collapse; this plays a similar role as our $W_2$ loss. Therefore, the marginal effect of the additional $W_2$ loss is reduced since Zero-CL already relieves the dimensional collapse to some extent.

---

> > ### Comment · Reviewer_KkL8 · 2023-11-22
> >
> > Thanks to your replies, I will keep the rating as I originally assigned to it.

---

### Official Review · Reviewer_RABe · 2023-10-31

**Soundness:** 3 good
**Presentation:** 4 excellent
**Contribution:** 3 good
**Rating:** 8
**Confidence:** 3

**Summary:**

The authors propose criteria for a uniformity loss during representation learning. This criteria is met through the use of the quadratic Wasserstein distance between learned representations and a uniform Gaussian distribution as a loss function, promoting uniformity without reduction in rank of the learned representation. Empirically this is shown to improve performance on the CIFAR-10/100 for a variety of models.

**Strengths:**

The paper is very well written, clearly stating the desired criteria and showing the proposed approach meets these criteria. The overall motivation behind the desired properties for a representation learning loss seem reasonable with a combination of theoretical justification and intuitive explanation/visualization provided.

Empirically, the results show an improvement by adding the proposed loss function. The authors do a nice job of evaluating the impact of this additional loss term for a variety of approaches.

**Weaknesses:**

The main weakness of this paper is that experiments are only done for the CIFAR-10/100 datasets. Given the proposed approach is claiming that dimensional collapse represents a fundamental issue with representation learning, the impact/relevance of this claim would be significantly strengthened by showing dimensional collapse is a fundamental issue and not necessarily a product of the CIFAR datasets.

On a related note it would be interesting to see the impact of the proposed metric on differing representation dimensions. In particular would the performance improvements still be observed if the representation dimension was doubled or tripled given that property 5 penalizes constant dimensions, intuitively making it seem that this would require careful selection of the representation dimension.

**Questions:**

Are there any results for differing representation dimensions?

(Not a negative comment, just hoping to get some insight into the behavior of the loss) Is there any intuitive reason as to why the top-5 accuracy decreases for MoCo and BarlowTwins on CIFAR-10 using the proposed loss even though the top-1 accuracy increases?

---

> ### Author Response · Authors · 2023-11-17
> **Response to reviewer RABe**
>
> **R 1.1 Are there any results for differing representation dimensions?**
>
> Generally speaking, an excess of dimensions can lead to redundancy; this is the basic rationale behind property 5, which penalizes constant dimensions. In line with Occam's Razor, we should avoid the unnecessary multiplication of entities. Although some extent of redundancy might be a common practice in modern machine learning and isn't always harmful- especially in the era of large language models - it is decidedly less efficient in contexts where conserving energy and resources is costly. Our preference leans strongly towards the Principle of Parsimony. On the other hand, insufficient dimensions result in a lack of learning capacity, rendering it challenging to grasp the inherent complexities of the data.
>
> We believe that our method follows a general principle that considers both efficiency and effectiveness in selecting dimension size; where carefully selecting the representation dimension is sometimes a tricky process.
>
> **R 1.2 Is there any intuitive reason as to why the top-5 accuracy decreases for MoCo and BarlowTwins on CIFAR-10 using the proposed loss even though the top-1 accuracy increases?**
>
> Thanks for pointing this out.  We find that incorporating an additional loss consistently enhances the performance on the CIFAR-10 dataset in terms of Acc@1 (Top-1 accuracy), but not in terms of Acc@5 (Top-5 accuracy). This phenomenon can be attributed to the near saturation of performance with respect to Acc@5. Notably, all models, whether baseline or those utilizing the additional W_2 loss, exhibit Acc@5 scores exceeding 99.6%. This implies that the correct label is almost always within the top 5 ranked classes. Given that CIFAR-10 encompasses only 10 classes in total, the performance metric of Acc@5 appears to be reaching a saturation point. In a 10-class classification task, the high saturation of Acc@5 as a performance metric makes it difficult to distinguish between the effectiveness of different models. Also, there exist 0.54% errors in the CIFAR-10 dataset; see Curtis G. Northcutt et al.
>
> G. Northcutt et.al.  Pervasive Label Errors in Test Sets Destabilize Machine Learning Benchmarks [https://arxiv.org/pdf/2103.14749.pdf](https://arxiv.org/pdf/2103.14749.pdf).

---

### Author Response · Authors · 2023-11-17
**General commention to all reviewers**

Thank you to all the reviewers for your valuable feedback. We sincerely appreciate your constructive comments, and we are grateful to reviewers *RABe* and *u9aL* for acknowledging the  motivation of our research. We also extend our gratitude to reviewer *KkL8* for highlighting that the proposed uniformity metric is more comprehensive. Additionally, we would like to thank reviewer *alG3* for recognizing the extensive empirical analysis and the overall quality of the paper.

In response to the reviewers’ feedback, we have made major revisions to the paper, summarized as follows:

- **Empirical comparisons on five properties** We have included empirical comparisons of the ICC, FCC, and FBC metrics in Section 5.2.
- **Experiments incorporating the $\mathcal{L_U}$ constraint** We have incorporated the existing uniformity loss $\mathcal{L_U}$ into the MoCo v2, BYOL, BarlowTwins, and Zero-CL frameworks, as presented in Table 2.
- **New theoretical results regarding Wasserstein distance between $Y_i$ and $\hat{Y}_i$** We have applied the $T_2$-inequality to prove that the quadratic Wasserstein distance between $Y_i$ and $\hat{Y}_i$ approaches zero as $m \to \infty$, as stated in Theorem 2.

Once again, we appreciate the reviewers’ thoughtful comments, and we have carefully addressed each of them in our revised paper.

---

### Meta-Review · Area_Chair_qCHy · 2023-12-08

**Metareview:**

Thanks for your submission to ICLR.

This paper explores uniformity metrics, and in particular discusses 5 properties that one should hold.  Then the authors demonstrate that the existing uniformity metric fails to have these properties, and further they propose a more effective uniformity metric that does meet the properties.

The reviews on this paper were somewhat mixed.  Some of the positives included: the paper is well written, there is good motivation of the approach, and the overall approach is novel and interesting.  On the negative side: there is limited empirical evidence provided, the sufficiency of the proposed criteria is not really established, and there is an assumption of Gaussanity used in the approach.

The authors did a nice job of responding to the criticisms of the reviewers.  It's worth noting also that the two negative reviews had generally minor concerns, and I think the authors addressed these concerns.  With that in mind I'm happy to recommend accepting the paper.

**Justification For Why Not Higher Score:**

This was a fairly borderline paper with two somewhat negative reviews; given the (mostly minor) concerns of these reviewers, the paper is probably not strong enough for a spotlight or oral.

**Justification For Why Not Lower Score:**

The concerns of the negative reviewers were mostly minor and were addressed satisfactorily in the rebuttal.

---

### Decision · Program_Chairs · 2024-01-16

Accept (poster)